# The bacterial toxin ExoU requires a host trafficking chaperone for transportation and to induce necrosis

Vincent Deruelle[1,3], Stéphanie Bouillot[1,4], Viviana Job[1,4], Emmanuel Taillebourg[2], Marie-Odile Fauvarque [2], Ina Attrée [1,4] & Philippe Huber [1,5✉]

*Pseudomonas aeruginosa* can cause nosocomial infections, especially in ventilated or cystic fibrosis patients. Highly pathogenic isolates express the phospholipase ExoU, an effector of the type III secretion system that acts on plasma membrane lipids, causing membrane rupture and host cell necrosis. Here, we use a genome-wide screen to discover that ExoU requires DNAJC5, a host chaperone, for its necrotic activity. DNAJC5 is known to participate in an unconventional secretory pathway for misfolded proteins involving anterograde vesicular trafficking. We show that DNAJC5-deficient human cells, or *Drosophila* flies knocked-down for the DNAJC5 orthologue, are largely resistant to ExoU-dependent virulence. ExoU colocalizes with DNAJC5-positive vesicles in the host cytoplasm. DNAJC5 mutations preventing vesicle trafficking (previously identified in adult neuronal ceroid lipofuscinosis, a human congenital disease) inhibit ExoU-dependent cell lysis. Our results suggest that, once injected into the host cytoplasm, ExoU docks to DNAJC5-positive secretory vesicles to reach the plasma membrane, where it can exert its phospholipase activity

[1] Université Grenoble-Alpes, CEA, INSERM, CNRS, Unité de Biologie Cellulaire et Infection, Grenoble, France. [2] Université Grenoble-Alpes, CEA, INSERM, CNRS, Unité de Biologie à Grande Echelle, Grenoble, France. [3] Present address: Unité de Biochimie des Interactions macromoléculaires, Département de Biologie Structurale et Chimie, CNRS UMR 3528, Institut Pasteur, Paris, France. [4] Present address: Université Grenoble-Alpes, CNRS, CEA, IBS, Grenoble, France. [5] Present address: Center for Immunology of Viral, Auto-immune, Hematological and Bacterial diseases (IMVA-HB/IDMIT), Université Paris-Saclay, Inserm, CEA, Fontenay-aux-Roses, France. ✉email: phuber@cea.fr

In most instances, bacterial toxins require one or more host factors to exert their toxicity. These factors can be receptors, binding partners inducing structural modifications or even entire host cellular pathways that are hijacked for bacterial toxicity purposes. The requirement for host-cell mechanisms protects bacteria from self-toxicity and takes advantage of efficient molecular mechanisms developed by eukaryotic cells to alter cellular functions. This rule applies to the toxins secreted by *Pseudomonas aeruginosa*, a Gram-negative opportunistic pathogen.

*P. aeruginosa* is a leading cause of severe nosocomial infections. It is a causative agent of pneumonia, urinary tract infections, bacteraemia, abscesses, as well as burn and eye infections. *P. aeruginosa* infections are frequent in ventilated and cystic fibrosis patients, and have a particularly high fatality rate following infection in these conditions[1–3]. The high mortality rate recorded is also due to acquired resistance to antibiotics, which is a major issue in the clinical management of *P. aeruginosa* infections[4–6].

*P. aeruginosa* uses a multi-target strategy to infect host cells, employing a combination of virulence factors. One of these factors is the type 3 secretion system (T3SS), the effectors of which are known to be the most potent toxins in acute *P. aeruginosa* infections[2,7]. The T3SS consists of a syringe-like apparatus which injects toxins into the cytosol of host cells. Four effectors have been identified: ExoU, ExoS, ExoT and ExoY, having their cognate co-activation host factors: phosphatidylinositol-4,5-bisphosphate [PI(4,5)P2] and ubiquitin for ExoU, 14-3-3 adaptor protein for ExoS and ExoT, and filamentous actin for ExoY[8–14].

ExoU and ExoS are mutually exclusively expressed in clinical strains[2]. ExoU-positive bacteria represent 28–48% of *P. aeruginosa* clinical isolates, and are found in the most severe pathological cases and produce the most dramatic lesions[2,7,15]. Furthermore, ExoU-positive strains have been associated with increased multidrug resistance in several clinical studies[16–19].

ExoU is a phospholipase A2 (PLA2) inducing plasma membrane rupture and rapid cell necrosis[20,21]. Its activity is enhanced by binding to ubiquitin and to PI(4,5)P2, a lipid present in the inner leaflet of the plasma membrane[10–14]. However, several aspects of ExoU activation and trafficking in host cells remain elusive. Here, we search for other host factors required for full ExoU toxicity using a genome-wide screening approach and we identify DNAJC5 as a necessary cofactor for its trafficking in host cells.

## Results

**ExoU requires DNAJC5 for host-cell lysis**. To identify host genes involved in ExoU cytotoxicity, we performed a genetic screen using CRISPR-Cas9 technology. A549 pneumocytic cells were transduced with a lentiviral library of guide-RNAs (gRNAs), targeting 18,053 genes (four gRNAs per gene). The cells were subjected to three rounds of infection with the *P. aeruginosa* strain PA14, known to induce cell necrosis via ExoU secretion (Fig. 1a). Each infection round was stopped by the addition of antibiotics after 90 min of infection, and each round of infection resulted in approximately 70% of cell death. This experimental design aimed at selecting resistant cells to ExoU-induced necrosis putatively carrying a mutated human gene required for ExoU necrotizing activity. The gRNA sequences in surviving A549 cells were identified by next-generation sequencing and the number of reads for each gRNA was compared to the number of reads in the uninfected library. Three independent replicates were performed and a statistical analysis revealed a significant enrichment for gRNAs targeting only one gene: the gene encoding DNAJC5 (also known as cysteine string protein α; CSPα)(Fig. 1b).

DNAJC5 is a ubiquitous cytoplasmic protein located at the surface of late endosomes (LEs). It functions as a co-chaperone in association with Hsc70 or Hsp70, which play a central role in protein homeostasis[22–24]. DNAJC5 is also required for an unconventional protein secretion pathway[25,26], recently described as Misfolding-Associated Protein Secretion (MAPS)[27]. In this process, misfolded cytosolic proteins are translocated into DNAJC5+ LEs near the endoplasmic reticulum, which are then transported to the plasma membrane[28,29]. Eventually, fusion of the vesicles with the plasma membrane allows the elimination of misfolded proteins directly into the extracellular milieu[30]. Alternatively, the vesicles can produce exosomes containing the misfolded proteins in the extracellular milieu[31].

To confirm the role of DNAJC5 in ExoU-dependent cytotoxicity, we generated independent DNAJC5$^{-/-}$ A549 cells using CRISPR-Cas9 technology and one of the gRNA targeting *DNAJC5* in the library (Supplementary Fig. 1a). A clonal population was selected, in which four bases from the coding sequence of both *DNAJC5* alleles were deleted and leading to the expression of an aberrant protein (Supplementary Fig. 1a, b). The DNAJC5$^{-/-}$ cells were subjected to a cytotoxic test, after infection with PA14 in the presence of propidium iodide (PI) to detect necrotic cells. PI incorporation was monitored by automated time-lapse microscopy (Fig. 1c). The proportion of native A549 cells exhibiting a necrotic phenotype increased with time and reached >75% at 5 h post-infection (pi); in contrast, DNAJC5$^{-/-}$ cells exhibited no PI incorporation. PA14 lytic capacity was restored when *DNAJC5* expression was rescued in DNAJC5$^{-/-}$ cells (DNAJC5$^{-/-}$::DNAJC5)(Fig. 1d). Similar experiments were performed with the clinical strain PP34, isolated from bacteraemia and secreting high amounts of ExoU. In these experiments, some necrosis (18%) was observed at late time points in DNAJC5$^{-/-}$ cells, while 100% of A549 cells were necrotic (Fig. 1e). As with PA14, PP34 infection of DNAJC5$^{-/-}$::DNAJC5 cells restored a full ExoU cytotoxicity (Fig. 1f). As control, we performed the same experiments with the PA14ΔexoU and PP34ΔexoU strains (Supplementary Fig. 2a, b). No PI incorporation was observed upon infection with these strains, confirming that the necrotizing activity of PA14 and PP34 requires ExoU. Furthermore, the fact that PP34 was partially toxic in DNAJC5$^{-/-}$ cells, while the *exoU* mutant strain was not, indicates that PP34-secreted ExoU can partially overcome the absence of DNAJC5 to exert its toxicity. To monitor cell necrosis with a different assay, we measured lactate dehydrogenase (LDH) release in supernatants of cells infected with PA14 or PP34 (Supplementary Fig. 2c). These data further establish the contribution of DNAJC5 in ExoU toxicity.

To determine whether DNAJC5 contributes directly to toxin activity, or is required for T3SS-dependent injection, we infected cells with bacteria secreting ExoU fused to β-lactamase (ExoU-Bla). This reporter system was previously used to monitor ExoU delivery into cells[32]. Host cells were pre-loaded with a fluorescent substrate of Bla (CCF2) used for fluorescence resonance energy transfer (FRET) experiments. Uncleaved CCF2 produces a green fluorescence, whereas the cleaved CCF2 emits a blue fluorescence. Infection of A549 or DNAJC5$^{-/-}$ cells with ExoU-Bla-secreting bacteria produced similar ratios of blue/green fluorescence (Fig. 1g), indicating that the absence of DNAJC5 did not alter T3SS injection per se.

As a complementary demonstration that DNAJC5 is required for ExoU toxicity, we used a DNAJC5 inhibitor. Quercetin is a natural product inducing DNAJC5 dimerization at high concentrations, leading to its inactivation[33]. Upon application of quercetin to cell cultures, a dose-dependent inhibition of ExoU-induced A549 cell lysis was observed (Fig. 1h), confirming the essential role of DNAJC5 in ExoU intoxication.

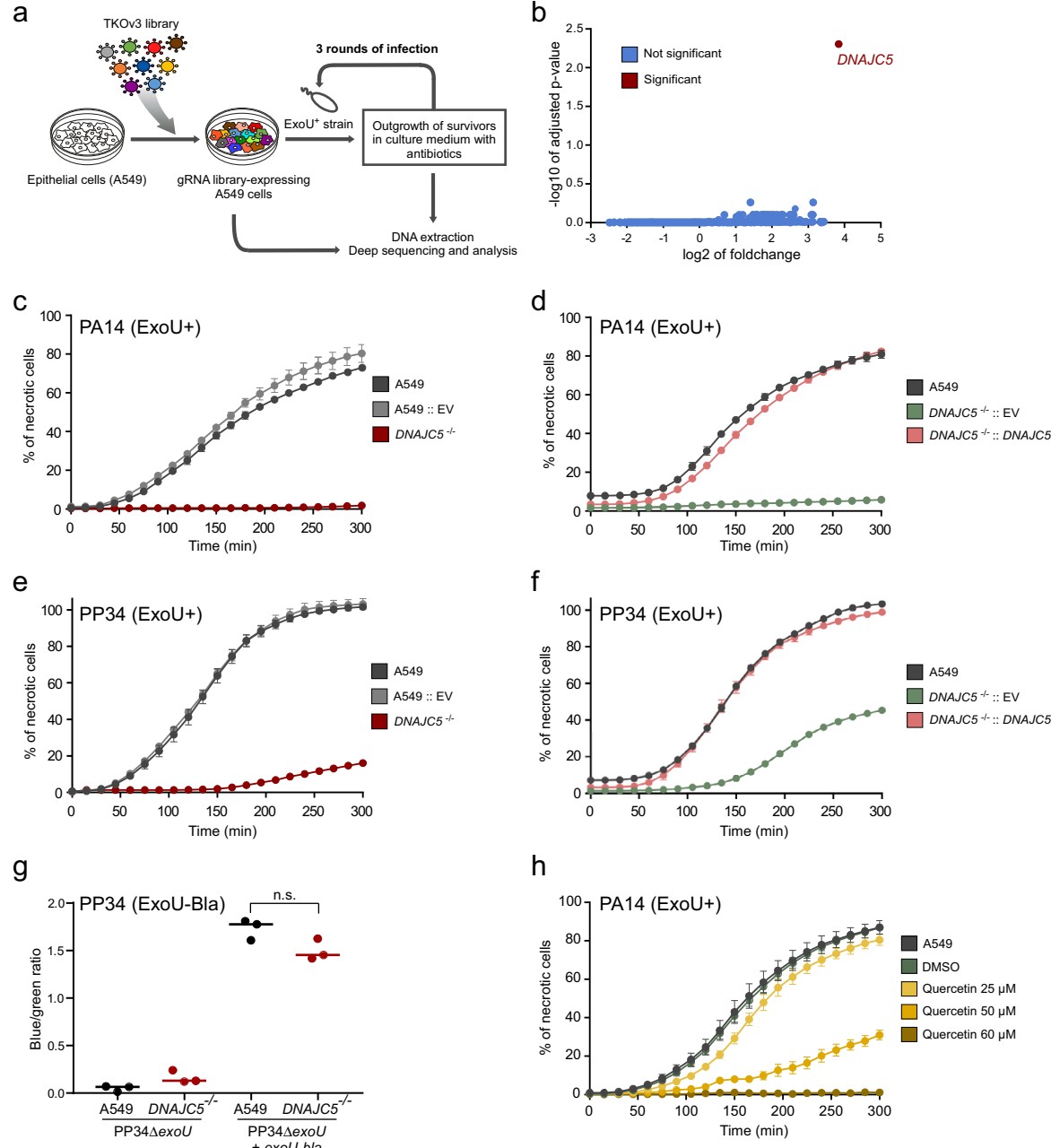

**Fig. 1 DNAJC5 is required for ExoU cytotoxicity. a** Screening process to identify host genes required for ExoU toxicity. A gRNA library (TKOv3, four gRNAs per gene) was constructed for A549 human epithelial cells. Cells were subjected to three 90-min rounds of infection with the ExoU+ PA14 strain in triplicates. Infection was stopped by washing and adding antibiotics. DNA, corresponding to gRNAs from resistant cells and from the uninfected library, were then amplified by PCR and submitted to deep sequencing. **b** Analysis of sequencing data. Statistical analysis of gRNA abundance in infected vs uninfected conditions. gRNAs targeting the DNAJC5 gene were the only ones significantly enriched in the screen. **c** Cytotoxicity assay. A clonal population of DNAJC5$^{-/-}$ A549 cells, native A549 cells or cells transfected with an empty vector (EV) were infected in triplicates with PA14 and necrosis was monitored by propidium iodide incorporation, recorded by time-lapse microscopy. Results are represented as the mean percentage (±SD) of necrotic cells. **d** Cytotoxicity assay with A549 native cells. DNAJC5$^{-/-}$ cells complemented with DNAJC5 (DNAJC5$^{-/-}$::DNAJC5), or the mock-complemented control, were infected with PA14. Results ($n = 4$ replicates) are represented as the mean percentage (±SD) of necrotic cells. **e**, **f** Cytotoxicity assays and representation are similar to (**c**,**d**), but cells were infected with the ExoU+ PP34 strain ($n = 3$ and 4 for (**e**) and (**f**), respectively). **g** T3SS-dependent injection of ExoU in A549 epithelial cells. A549 or DNAJC5$^{-/-}$ cells were infected with PP34Δ*exoU* expressing ExoU$^{S142A}$ fused to the β-lactamase (Bla) or infected with uncomplemented PP34Δ*exoU*. Cells were loaded with CCF2, a fluorescent substrate for Bla, which shifts from green to blue fluorescence upon processing by the enzyme. Fluorescence was measured at 4 hpi on both channels and results are expressed as a blue/green ratio ($n = 3$; bar: median). Statistical differences between data in A549 and DNAJC5$^{-/-}$ cells infected with the *exoU-bla* strain were calculated with a two-sided Student's test, and were not significant (n.s.). **h** Cytotoxicity assay. Increasing concentrations of quercetin were added to A549 cells in the presence of PA14 (ExoU+) at an MOI of 10. Necrosis was monitored by PI incorporation and recorded by time-lapse microscopy. Results ($n = 8$ replicates) are represented as the mean percentage (±SD) of necrotic cells. Source data are provided as a Source Data file.

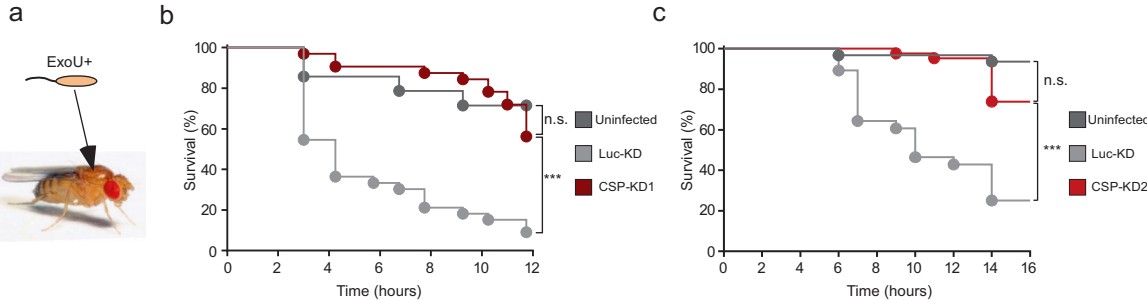

**Fig. 2 ExoU toxicity in *Drosophila* requires CSP (DNAJC5 orthologue). a** *Drosophila* were infected with PP34 (ExoU+) by pricking the thorax with a thin needle dipped in bacterial suspension. **b** Flies expressing RNAi transgenes targeting either the firefly *luciferase* gene (Luc-KD, n = 33) or the *Csp* gene (CSP-KD1, n = 40) were infected with PP34. **c** Similar experiment with Luc-KD (n = 28) and CSP-KD2 (n = 51) infected with PP34. Mock-infected CSP-KD1 (**b**, n = 30) or CSP-KD2 (**c**, n = 47) flies were included as negative control (uninfected). Fly survival was recorded and data are represented as Kaplan–Meyer curves. Multiple comparisons tests (LogRank) gave a p-value of 0.0001 for (**b**, **c**). Simple LogRank comparison tests were performed: n.s., non-significant; ***p < 0.0001. Source data are provided as a Source Data file.

We next examined whether DNAJC5 also contributes to the activity of other toxins delivered by *P. aeruginosa*. To do so, we first performed a cytotoxicity assay using bacteria CHAΔ*exoT* secreting ExoS (and not ExoU or ExoT) through the T3SS. The toxic activity of ExoS results in dismantling of the actin cytoskeleton, hence provoking cell rounding[2]. The similar kinetic profiles recorded for the two cell lines (Supplementary Fig. 3a, b), indicate that the action of ExoS in host cells does not require DNAJC5. This result also further confirmed that T3SS injection is unaffected in DNAJC5$^{-/-}$ cells.

We subsequently assayed how DNAJC5 deficiency affected toxicity of the pore-forming toxin ExlA, secreted by *P. aeruginosa* strains lacking T3SS[34]. Like ExoU, ExlA is a necrotizing toxin. Identical intoxication curves were recorded for A549 and DNAJC5$^{-/-}$ cells (Supplementary Fig. 3c), indicating that DNAJC5 is not involved in ExlA-dependent cell lysis. Likewise, DNAJC5 seems to be specifically required for ExoU necrotizing activity in host cells.

Taken together, these results demonstrate that DNAJC5 is specifically required for ExoU toxicity in host cells.

**Lack of DNAJC5 decreases the virulence of ExoU-positive *P. aeruginosa*'s strains in vivo.** DNAJC5 is an evolutionary conserved protein. Animals in which the *DNAJC5* orthologue gene was inactivated (mice, *Caenorhabditis elegans* and *Drosophila melanogaster*) all rapidly died after birth from neurological disorders[35–37]. These models are consequently unsuitable for use in infection assays. To overcome this lack of in vivo model, we generated *Drosophila* in which the *DNAJC5* orthologue *Cystein string protein* (*Csp*) was conditionally knocked-down (KD) at the adult stage by expressing a silencing long double-strand RNA (dsRNA) or a short hairpin RNA (shRNA) in two independent fly lines (generation of the CSP-KD flies is shown in Supplementary Fig. 4a, b). Flies were infected by pricking the thorax with a thin needle previously dipped into a bacterial suspension[38] (Fig. 2a). This infection model can be used to measure the impact of ExoU on fly death, as shown by the difference in survival curves following infection with a wild-type strain of *P. aeruginosa* expressing ExoU (PP34) and its isogenic mutant (PP34Δ*exoU*) (Supplementary Fig. 4c). Survival curves for control and CSP-KD *Drosophila* infected with PP34 were strikingly different (Fig. 2b, c), whereas the survival curves for infected CSP-KD and *Drosophila* mock-infected with PBS were not statistically different. The CFU numbers were not significantly different between flies infected with PP34 and PP34Δ*exoU* for the three backgrounds (Luc-KD, CSP-KD1 and CSP-KD2) at 2 and 6 hpi (except for CSP-KD1 flies at 2 hpi) pointing to a major role of ExoU in fly

death, rather than a difference in bacterial growth (Supplementary Fig. 4d).

These results show that CSP-KD flies are more resistant to ExoU-induced death than control flies. Although one cannot formally exclude an unlikely developmental effect (see Materials and Methods for detail), these data indicate that DNAJC5/CSP is required for full ExoU-dependent *P. aeruginosa* virulence in vivo.

Interestingly, the survival curves of *Drosophila* infected with PA14 or PA14Δ*exoU* were not significantly different (Supplementary Fig. 4e), suggesting that PA14 lethal effect in *Drosophila* is mostly mediated by virulence factors other than ExoU. Therefore, PA14 was not used in this in vivo assay to examine the role of CSP in ExoU toxicity.

**ExoU partly localizes in DNAJC5-positive vesicles.** To determine the localization of ExoU in infected cells, we used a *P. aeruginosa* strain CHAΔ*exoSexoT::exoU*$^{S142A}$ (hereafter CHA-*exoU*$^{S142A}$) that secretes a catalytically inactive non-lytic mutant, ExoU$^{S142A}$[39]. Cells were infected with this strain, and soluble and membrane fractions were prepared from lysates of A549 infected cells. ExoU was mainly detected in the membrane fractions, with only trace amounts present in soluble fractions (Fig. 3a). This distribution indicates that following injection, ExoU binds to membranes rather than remaining free in the cytosol. Equivalent amounts of ExoU were detected in DNAJC5$^{-/-}$ and DNAJC5$^{-/-}$::DNAJC5 membrane fractions, demonstrating that DNAJC5 is not required for ExoU docking to membranes. To gain further insight into the subcellular localization of ExoU, we performed immunofluorescence experiments and observed cells by confocal microscopy. In infected A549 and DNAJC5$^{-/-}$ cells, ExoU displayed similar particulate labeling in the cytoplasm of both cell types (Fig. 3b). Thus, once ExoU is injected into cells, it binds to specific cytoplasmic structures in a DNAJC5-independent manner.

To determine whether ExoU colocalizes with DNAJC5 in these intracellular structures, ExoU and DNAJC5-Flag were labeled in DNAJC5$^{-/-}$::DNAJC5 cells infected with CHA-*exoU*$^{S142A}$. Most DNAJC5-FLAG labeling was observed at the cellular periphery, associated with elongated vesicles, and at cell-cell junctions (Fig. 3c). Round DNAJC5+ vesicles were also present in the perinuclear region. These perinuclear DNAJC5+ vesicles were also positive for the lysosome/late endosome marker Lamp2 (Supplementary Fig. 5), as previously reported for Cos7 cells[28], indicating that these vesicles are LEs. ExoU colocalized with DNAJC5 in both elongated and round vesicles, as well as at cell-cell junctions, in the top and middle parts of the cell. However, in the lower part of the cell, the two labels were partly dissociated

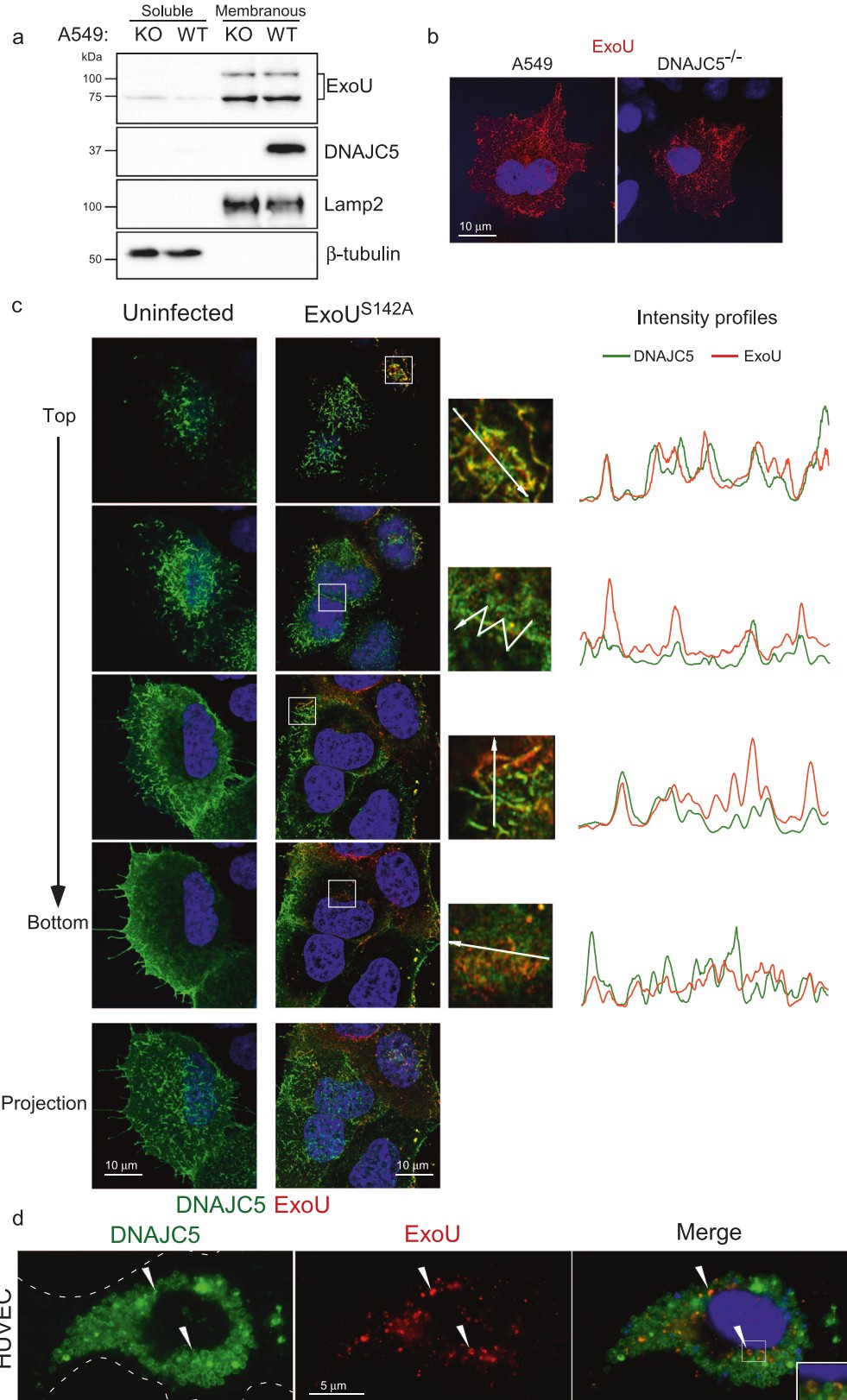

(Fig. 3c). Interestingly, the presence of ExoU had no impact on the subcellular localization of DNAJC5+ vesicles.

To allow a more detailed examination of DNAJC5/ExoU colocalization, we performed a similar experiment with endothelial cells (HUVECs), which have larger, mostly perinuclear, DNAJC5 + LEs. In these cells, both DNAJC5-GFP and ExoU were localized at the vesicle's limiting membrane and were not intraluminal (Fig. 3d), confirming that ExoU is membrane-associated.

ExoU has been shown to be partially ubiquitylated at K178, and its ubiquitylated form was found preferentially associated with early endosomes[39,40] (Fig. 3a). Although, the K178R

**Fig. 3 ExoU localizes in DNAJC5-positive vesicles. a** Fractionation of DNAJC5$^{-/-}$ (KO) and DNAJC5$^{-/-}$::DNAJC5 (WT) cells infected with a *P. aeruginosa*'s strain (CHA-*exoU*$^{S142A}$) secreting a catalytically inactive ExoU mutant through its T3SS. Cells were harvested at 4 hpi and their soluble and membrane fractions were prepared. Western blots were performed on fractions using anti-ExoU, DNAJC5, Lamp2 (a late endosome-lysosome marker) and β-tubulin (a cytosolic marker) antibodies. The higher molecular weight bands revealed by the ExoU antibody represent the ubiquitinylated form of ExoU. Source data are provided as a Source Data file. **b** ExoU immunofluorescence staining of A549 and DNAJC5$^{-/-}$ cells infected with CHA-*exoU*$^{S142A}$. A single representative z-section is shown. **c left** DNAJC5-FLAG immunofluorescence signals (green) in uninfected DNAJC5$^{-/-}$::DNAJC5 cells. Four z-sections obtained by confocal microscopy are shown from top to bottom. A z-projection is also shown below. Nuclei were counterstained in blue. **c right** DNAJC5-FLAG (green) and ExoU (red) immunofluorescence signals in DNAJC5$^{-/-}$::DNAJC5 cells infected with CHA-*exoU*$^{S142A}$. As for uninfected cells, four z-sections and a z-projection are shown. For each section, a region was enlarged and an arrow was drawn (represented on the right) to establish an intensity profile for both green and red fluorescences, as shown. **d** DNAJC5-GFP and ExoU localizations in transfected HUVEC infected with CHA-*exoU*$^{S142A}$ on a wide-field microscopy image. Arrowheads show colocalization of both markers. The insert is an enlargement of the merged image, showing DNAJC5 and ExoU localization at the vesicle's membrane.

mutation did not affect ExoU necrotizing activity[40], we examined whether the K178R mutation would alter ExoU localization at DNAJC5+ vesicles (Supplementary Fig. 6). ExoU$^{S142A-K178R}$ was found similarly associated with DNAJC5+ vesicles, suggesting that ExoU ubiquitylation is not required for its interaction with these vesicles.

**Hsp70 and Hsc70 chaperones are dispensable for ExoU toxicity.** As the T3SS does not accommodate folded proteins[41], ExoU is probably delivered unfolded into host cells. Therefore, we reasoned that ExoU might need the chaperone activity of the DNAJC5-Hsc70/Hsp70 complex to recover its catalytic activity.

Three domains have been identified in DNAJC5 (Fig. 4a). A J domain, present in all DNAJ proteins, a cysteine string domain containing 14 cysteines and a C-terminal domain[42]. The J domain interacts with Hsc70/Hsp70 and enhances the ATPase activity of these chaperones[43]. Palmitoylation of cysteines in the central domain allows initial DNAJC5 anchoring at the surface of endosomes[44]. The C-terminal domain has been shown to associate with various proteins, including the vesicle-associated membrane protein (VAMP), a SNARE protein involved in the fusion of vesicles with the plasma membrane[45,46].

To determine whether the co-chaperoning role of DNAJC5 is linked to ExoU toxicity, we produced cells carrying several mutations to disrupt interactions between DNAJC5 and Hsp70/Hsc70, and infected them with ExoU+ *P. aeruginosa*. First, we complemented DNAJC5$^{-/-}$ cells with DNAJC5$^{H43Q}$, as this mutation was previously shown to disrupt DNAJC5-Hsc70/Hsp70 interaction[47]. DNAJC5$^{-/-}$::DNAJC5$^{H43Q}$ cells displayed higher sensitivity to *P. aeruginosa*-induced cell lysis than DNAJC5$^{-/-}$::DNAJC5 (Fig. 4b), indicating that DNAJC5 interaction with Hsc70/Hsp70 is not required for ExoU activation. To better understand why the H43Q mutation lead to higher sensitivity to *P. aeruginosa*, we performed toxicity assays on DNAJC5$^{-/-}$::DNAJC5$^{H43Q}$ cells using two isogenic strains devoid of ExoU. PA14Δ*exoU* did not induce necrosis in DNAJC5$^{-/-}$::DNAJC5$^{H43Q}$ cells, while PP34Δ*exoU* did (Supplementary Fig. 2a, b); no PI incorporation was detected when cells were incubated without bacteria (not shown). Thus, cells carrying the H43Q mutation are more sensitive to *P. aeruginosa*'s factors other than ExoU. This secondary toxicity likely explains the increased necrotizing activity of ExoU in DNAJC5$^{-/-}$::DNAJC5$^{H43Q}$ cells when infected with ExoU-expressing strains (Fig. 4b).

Shirafuji et al have reported that phosphorylation of two serines (S10 and S34) enhanced the co-chaperone activity of DNAJC5[48]. Furthermore, they showed that replacing the two serines by alanines (DNAJC5$^{S10A-S34A}$) reduced DNAJC5 interaction with Hsp70. Therefore, we used DNAJC5$^{-/-}$ cells complemented with DNAJC5 carrying both mutations in a cytotoxicity assay. ExoU displayed similar toxicity in

DNAJC5$^{-/-}$::DNAJC5 and DNAJC5$^{-/-}$::DNAJC5$^{S10A-S34A}$ cells (Fig. 4b), providing further evidence that DNAJC5's co-chaperone activity is not required for ExoU activation. Interestingly, the H43Q and S10A-S34A mutations had no effect on subcellular localization of DNAJC5, nor on its colocalization with ExoU and Lamp2 (Fig. 4c and Supplementary Figs. 5, 7).

To definitively investigate the role played by Hsp70/Hsc70 in ExoU toxicity, we knocked-down each of these two proteins in A549 cells using siRNAs (Fig. 4d). Both KD cell lines were sensitive to PA14-induced lysis with a profile similar to native cells (Fig. 4e).

Based on these results, and despite the proven role of Hsc70/Hsp70 in protein exocytosis in MAPS[26], these chaperones appear not to be required for ExoU intoxication.

**DNAJC5-positive vesicles escort ExoU to the plasma membrane.** Having eliminated its co-chaperone role, we next investigated whether the trafficking activity of DNAJC5 was required for ExoU-dependent necrosis.

Adult neuronal ceroid lipofuscinosis is a neurodegenerative disease caused by mutations in *DNAJC5*[49–51]. Two mutations have been reported in patients: L115R and ΔL116 (stars in Fig. 4a). These mutations form protein oligomers[52,53] that impair the trafficking of DNAJC5+ vesicles towards the plasma membrane, causing the accumulation of misfolded proteins in cells, and leading to progressive neuronal dysfunction[52]. When expressed in DNAJC5$^{-/-}$ cells, DNAJC5$^{L115R}$ and DNAJC5$^{ΔL116}$ localized in intracellular vesicles, but not near or at the plasma membrane (Supplementary Fig. 7), a feature previously reported in PC12 neuroblastic cells[52]. Furthermore, these DNAJC5+ vesicles were only partially labeled with Lamp2, which is associated with other perinuclear vesicles (Supplementary Fig. 5). In both DNAJC5$^{-/-}$::DNAJC5$^{L115R}$ and DNAJC5$^{-/-}$::DNAJC5$^{ΔL116}$, ExoU colocalized with DNAJC5-FLAG (Fig. 5a, b). However, ExoU toxicity was severely diminished, recovering only minimally with DNAJC5$^{L115R}$ or moderately with DNAJC5$^{ΔL116}$ in these complemented cell lines (Fig. 5c).

To confirm these results, we produced another DNAJC5 mutant, lacking the J domain and also forming protein oligomers[53], which has been hypothesized to alter DNAJC5 transportation function. DNAJC5$^{ΔJdomain}$ similarly lead to restricted DNAJC5 localization in the perinuclear region, with some enlarged vesicles (Supplementary Figs. 5, 7), possibly due to protein accumulation in the lumen, where ExoU and Lamp2 colocalized (Fig. 5d). As with the pathological mutations L115R and ΔL116, ExoU toxicity was dramatically reduced in DNAJC5$^{-/-}$::DNAJC5$^{ΔJdomain}$ cells (Fig. 5e).

Based on these results, we concluded that DNAJC5 mutations affecting vesicle trafficking to the plasma membrane block ExoU-driven cell necrosis.

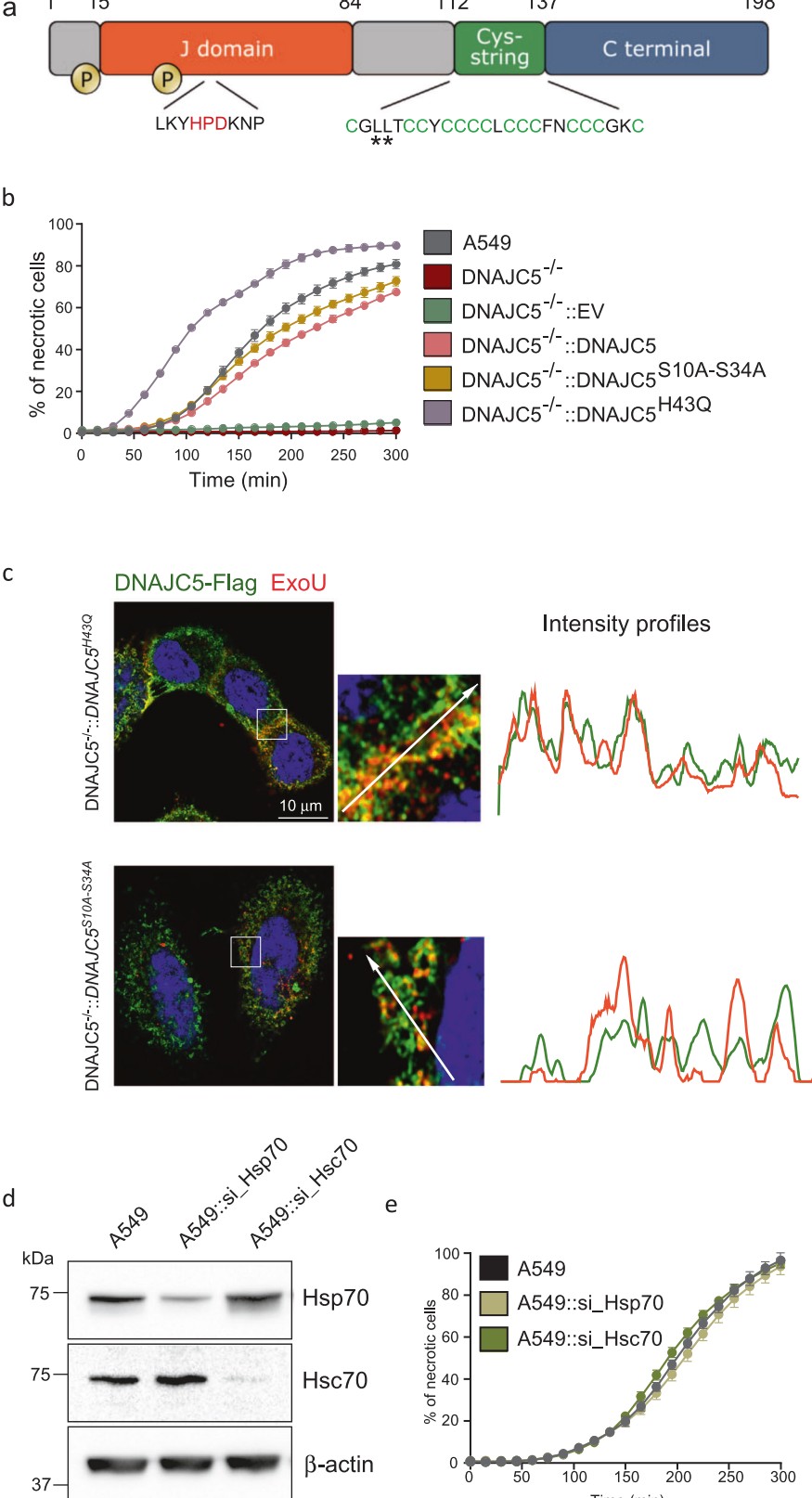

**ExoU phospholipase activity is independent of DNAJC5.** To assess whether DNAJC5 contributes to the activation of ExoU enzymatic activity per se, we expressed and purified recombinant ExoU and examined its catalytic activity in a PLA2 assay, in the presence of membrane or soluble fractions prepared from uninfected DNAJC5$^{-/-}$ or DNAJC5$^{-/-}$::DNAJC5 cells. Only the membrane fractions enhanced ExoU phospholipase activity (Fig. 6), suggesting that a membranous component, probably PI (4,5)P2 or other lipids, activates ExoU catalytic activity, as previously reported[14]. Importantly, no significant difference was detected when ExoU was incubated with membrane fractions from either DNAJC5$^{-/-}$ or DNAJC5$^{-/-}$::DNAJC5 cells, showing

**Fig. 4 DNAJC5 co-chaperone activity is dispensable for ExoU toxicity. a** Domain organization of the human DNAJC5 protein. The following locations are highlighted: the two phosphorylation sites (serine 10 and serine 34), the HPD motif for Hsc70/Hsp70 binding, and the cysteine-rich region containing the leucines L115 and L116, mutated in adult neuronal ceroid lipofuscinosis patients (stars). **b** Effect of DNAJC5 mutations, H43Q and S10A-S34A, on ExoU toxicity. A549 cells, as well as DNAJC5$^{-/-}$ cells complemented with either DNAJC5, DNAJC5$^{H43Q}$, DNAJC5$^{S10A-S34A}$ or the empty vector (EV) were subjected to an infection assay with PA14. Data are shown as the mean ± SD. $N = 6$ fields per condition. **c** DNAJC5-FLAG (green) and ExoU (red) immunofluorescence signals for DNAJC5$^{-/-}$::DNAJC5$^{H43Q}$ and DNAJC5$^{-/-}$::DNAJC5$^{S10A-S34A}$ cells infected with CHA-exoU$^{S142A}$ strain, which secretes a catalytically inactive ExoU mutant. One z-section is shown. For each section, a region was enlarged and an arrow was drawn (represented on the right) to establish the intensity profiles for both green and red fluorescences. **d**, **e**. Effect of decreased Hsc70 and Hsp70 expression on ExoU toxicity. A549 cells were transfected with siRNAs for Hsp70 (si_Hsp70) or Hsc70 (si_Hsc70) to knock down their expression. Knockdown was monitored by Western blot (**d**). KD cells were subjected to an infection assay (**e**)($n = 5$), in the same conditions as in (**b**). Source data are provided as a Source Data file.

that DNAJC5 is not directly involved in ExoU phospholipase activity.

## Discussion

The aim of this study was to identify host factors required for full ExoU toxicity using a genome-wide screening approach. Our results demonstrate that the host chaperone DNAJC5 is required for the toxic activity of ExoU both in human cells and in *Drosophila*. In the bacterial cytoplasm, ExoU forms a complex with its cognate chaperone, SpcU, from which it dissociates prior to travel, probably unfolded, through the injectisome[54]. Once injected into the host cell, ExoU partly colocalizes with DNAJC5+ vesicles in the perinuclear zone (i.e., LEs) and at the cellular periphery. DNAJC5 and Lamp2 only colocalized in perinuclear vesicles, and Lamp2 labeling was lost when DNAJC5 + vesicles were exported to the cellular periphery (Supplementary Fig. 5). Similar patterns were reported in Cos7 cells with Lamp1, another late endosome/lysosome marker[29], indicating that peripheral DNAJC5+ vesicles cannot be considered strictly "late endosomes".

DNAJC5 has been linked to an unconventional secretion system (MAPS), both for the translocation of misfolded cytosolic proteins into the vesicle's lumen and for the transport of these vesicles to the plasma membrane. DNAJC5-dependent protein exocytosis uses one of the two processes: either the vesicle fuses to the plasma membrane, allowing the release of the vesicle's intraluminal content into the extracellular milieu[30], or exosomes are produced[31].

Unlike classical MAPS cargos, ExoU localized to the vesicle's limiting membrane (Fig. 3d), it is therefore probably not translocated into the vesicle's lumen, and remains at the external side. If our hypothesis is correct, this position would allow ExoU to target the inner leaflet of the plasma membrane once the vesicle reaches the cell's periphery (Fig. 7). Moreover, MAPS cargos need Hsp70/Hsc70 chaperones for secretion through this pathway[26], while ExoU transport is Hsp70/Hsc70-independent (Fig. 4). This lack of chaperone dependence is probably linked to its position on the external side of the vesicle membrane.

ExoU toxicity was altered by mutations known to alter DNAJC5 function in vesicle trafficking, i.e., L115R and ΔL116[42,50,52], as well as J-domain deletion. These results suggest that ExoU, once delivered into the host's cytoplasm, uses the DNAJC5-associated secretion machinery for transport to the plasma membrane. This result also suggests that patients with adult neuronal ceroid lipofuscinosis might be more resistant to infections caused by ExoU-expressing *P. aeruginosa* strains.

Although ExoU and DNAJC5 colocalized at the vesicle's limiting membrane, our attempts to identify a physical association between ExoU and DNAJC5 were unsuccessful. Furthermore, lack of DNAJC5 did not prevent ExoU from associating with membranes (Fig. 3a, b), confirming that ExoU binding to intracellular vesicles is DNAJC5-independent. Therefore, although DNAJC5 is an essential component of the ExoU transportation

pathway, it is not the receptor for ExoU at the vesicle's surface. Consequently, further studies will be required to identify the receptors allowing ExoU to dock to vesicle membranes.

The importance of ubiquitin binding to ExoU to stabilize the ExoU-membrane association and enhance its catalytic activity was demonstrated in previous studies[10,11,55,56]. It will now be interesting to examine whether ubiquitin binding also facilitates or strengthens the association between ExoU and DNAJC5 + vesicles, as it does at the plasma membrane.

Several bacterial toxins exploit vesicle trafficking for retrograde transport[57], including T3SS cytotoxins[58,59]. In particular, *P. aeruginosa*'s ExoS toxin moves to the perinuclear region in a microtubule- and dynamin-dependent process, suggesting involvement of the endocytic pathway[60]. We previously showed that ExoU partially colocalized in cytoplasmic vesicles with EEA1[39], a protein marker of early endosomes. It is thus possible that ExoU reaches the perinuclear region by an endocytic pathway like that exploited by ExoS, because vesicle transport is faster than free diffusion in the cytosol.

According to Deng et al.[60], ExoS recycles to the plasma membrane by an unknown process. Our results showing full ExoS toxicity in DNAJC5$^{-/-}$ cells (Supplementary Fig. 3a, b) demonstrate that ExoS and ExoU use distinct pathways to reach the plasma membrane.

Among several tested phospholipids, only PI(4,5)P2, localized at the plasma membrane, can induce a conformational change in ExoU structure, leading to ExoU oligomerization and activating its catalytic activity[12,14,61]. In agreement with this specific activation mechanism, our results show that DNAJC5-dependent vesicle trafficking is required to induce ExoU-dependent membrane disruption (Fig. 1c).

In conclusion, we have identified a trafficking system required for ExoU-dependent toxicity in host cells. Importantly, our initial screen for factors involved in this targeting identified only *DNAJC5*, suggesting that the protein it produces may be the Achilles' heel for this highly potent toxin. Inhibitors of DNAJC5 (with a better efficiency than quercetin) or the MAPS pathway could be of considerable interest for adjunct therapy to treat infections with ExoU+ *P. aeruginosa* strains.

## Methods

**Bacterial strains and plasmids.** The bacterial strains and plasmids used in this study are listed in Supplementary Table 1. Bacterial strains were grown in Luria-Bertani (LB) medium at 37 °C with vigorous shaking (300 rpm). When mentioned, antibiotics were added at the following concentrations: 100 μg mL$^{-1}$ ampicillin and 300 μg mL$^{-1}$ carbenicillin. For infection assays, overnight cultures were diluted to optical density (OD$_{600}$) of 0.1 and grown under agitation to reach OD$_{600}$ of 1.

For the construction of PP34Δ*exoU*, the internal nucleotide sequence of *exoU* from PP34 isolate[62] was amplified by PCR and cloned into pEX100T[63]. The gentamicin cassette was extracted from pUC-Gm and inserted into the unique *Eco*RI site of *exoU*. Then, the *exoU*-Gm fragment was introduced into the genome of *P. aeruginosa* PP34 by allelic exchange following standard triparental mating and *sacB* selection technique. The mutant was verified by PCR using primers ExoU-Gm_Fw and ExoU-Gm_Rev (Supplementary Table 2) and resequencing.

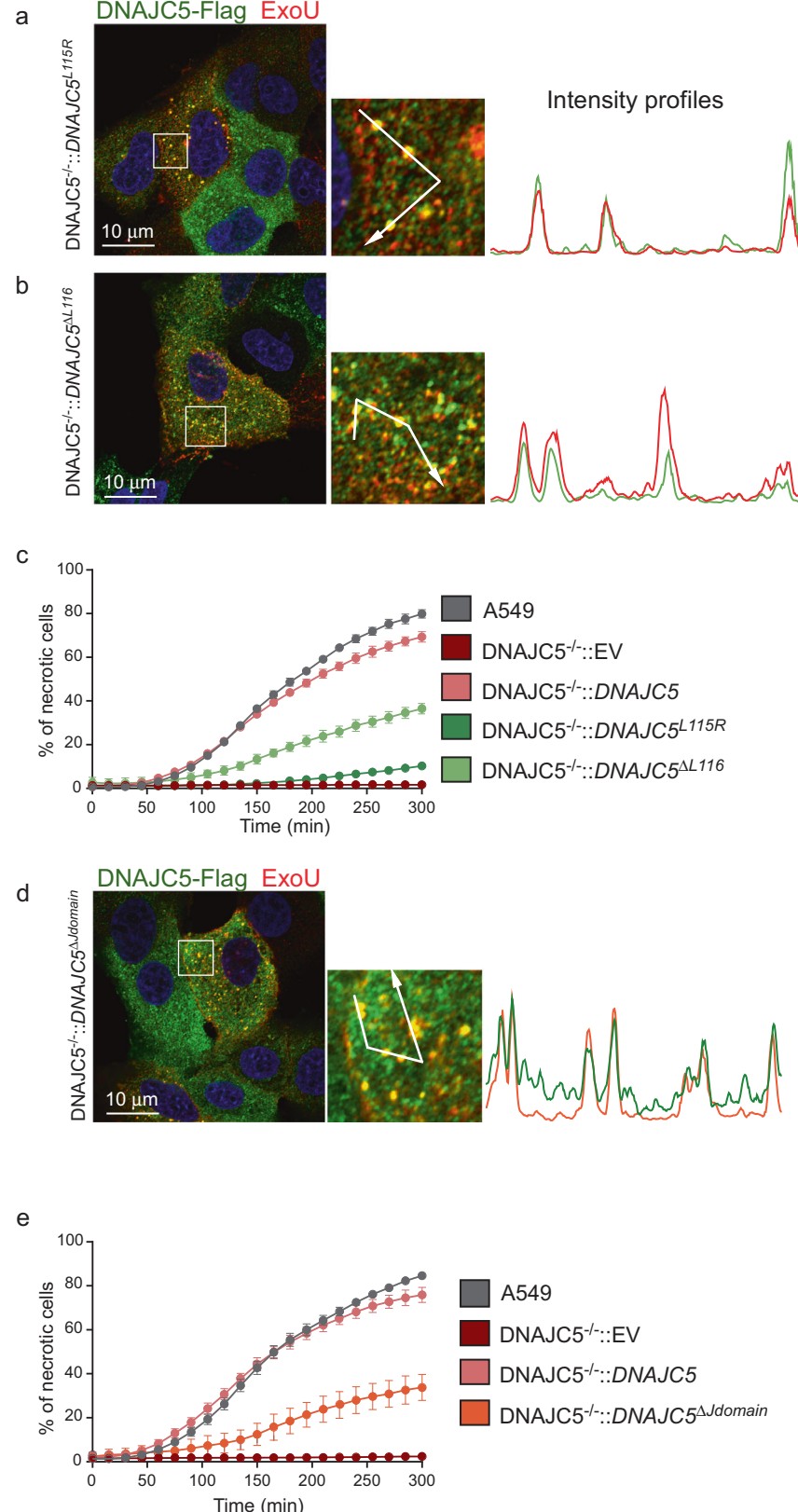

For *exoU-bla* fusion, the DNA fragment encompassing the *exoU* promotor and coding sequence was amplified by primers ExoU-BamHI and ExoU-XbaI (Supplementary Table 2) using pIA*exoU*^S142A*spcU* as template[39]. The PCR fragment was then transferred into *Bam*HI- and *Xba*I-digested pIA*exoS-bla* plasmids, where *exoS* was excised. The expression and secretion of ExoU-Bla were verified by immuno-blotting using anti-ExoU antibodies. The plasmids were introduced into PP34Δ*exoU* by transformation[64].

**Lentiviral production using TKOv3 library and MOI determination.** Toronto human knockout pooled library (TKOv3) was a gift from Jason Moffat and obtained from Addgene (#90294). It is a one-component library with guide-RNAs inserted in lentiCRISPRv2 backbone as well as the *Cas9* gene. This library contains four gRNAs targeting each of the 18,053 protein coding genes and control gRNAs targeting EGFP, LacZ and luciferase (71,090 total gRNAs). The gRNA library-expressing lentiviruses were produced as described in Moffat's lab protocol

**Fig. 5 Trafficking of DNAJC5-positive vesicles is required for ExoU toxicity. a** Localizations by immunofluorescence of ExoU (red) and DNAJC5-FLAG (green) in DNAJC5$^{-/-}$::DNAJC5$^{L115R}$ cells infected with CHA-$exoU^{S142A}$ on a z-section. A selected area was enlarged and an arrow was drawn, to establish the intensity profiles for both signals. **b** Immunofluorescence localizations of ExoU (red) and DNAJC5-FLAG (green) in DNAJC5$^{-/-}$::DNAJC5$^{\Delta L116}$ cells infected with CHA-$exoU^{S142A}$. Intensity profiles were obtained as above. **c** Cytotoxicity assay with DNAJC5$^{-/-}$::DNAJC5$^{L115R}$ and DNAJC5$^{-/-}$:: DNAJC5$^{\Delta L116}$ cells, alongside controls, infected with PA14. Data are shown as the mean ± SD. $N = 7$ fields per condition. **d** Immunofluorescence localizations of ExoU (red) and DNAJC5 (green) in DNAJC5$^{-/-}$::DNAJC5$^{\Delta Jdomain}$ cells infected with CHA-$exoU^{S142A}$. Intensity profiles were obtained as above. **e** Effect of J domain deletion. Control cells and DNAJC5$^{-/-}$ cells complemented with full-length DNAJC5, DNAJC5$^{\Delta Jdomain}$ or EV were infected with PA14, and cytotoxicity was recorded (mean ± SD; $n = 5$). Source data are provided as a Source Data file.

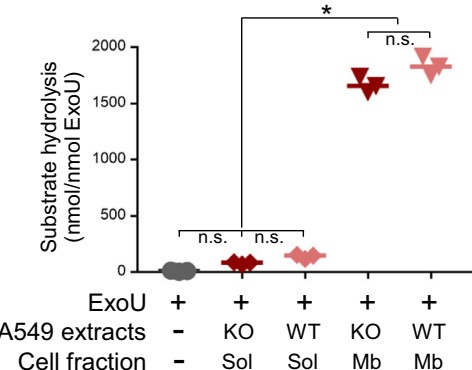

**Fig. 6 ExoU phospholipase activity is independent of DNAJC5.**
Phospholipase activity of purified ExoU (65 pmols) was measured in the presence of soluble (Sol) or membrane (Mb) fractions from uninfected DNAJC5$^{-/-}$ (KO) or DNAJC5$^{-/-}$::DNAJC5 (WT) cells. Experiments were performed in triplicates and incubated for 24 h. Data are expressed in nmoles of substrate hydrolyzed per nmoles of ExoU ($n = 3$; bar: median). Statistical differences were established by two-sided ANOVA ($p < 0.0001$), followed by Tukey's test (*$p < 0.0001$, n.s., non-significant). Source data are provided as a Source Data file.

(REV.20170404). The library was amplified in Lucigen Endura electrocompetent cells (#60242) to reach at least 200x colonies per guide-RNA and the library plasmid pool was purified using NucleoBond Xtra-Midi EF Kit (Macherey-Nagel, #740420.50). Then, lentiviruses were produced by transfection of HEK293T cells with the library plasmid pool. In brief, X-tremeGENE 9 DNA Transfection reagent (Roche, #06365787001) was diluted in Opti-MEM serum-free media. Following 5 min of incubation at room temperature, an appropriate mixture of plasmids was added in a 3:1 ratio of Transfection Reagent: DNA complex. This mixture of plasmids was composed of the library plasmid pool and the lentiviral packaging and envelope plasmids (psPAX2 and pMD2.G) at a 1:1:1 molar ratio (8 μg:4.8 μg:3.2 μg, respectively). The solution was then mixed and incubated at room temperature for 30 min. Thereafter, the transfection mix was added to 70–80% confluence HEK293T cells in a drop-wise manner and cells were incubated for 24 h. Following this incubation time, media were replaced by fresh DMEM, containing 6% of bovine serum albumin (BSA). The next day, the lentiviruses-containing media were harvested, centrifuged to pellet any packaging cells, and supernatants were stored at −80 °C. The lentiviral concentration was established by serial dilutions on A549 cells, as described below, treated or not treated with 2 μg mL$^{-1}$ puromycin. The virus volume that gave 30% survival with puromycin selection vs without puromycin was chosen for library construction, in order to limit the number of lentiviruses per cell.

**Construction of CRISPR-Cas9 library in A549 cells and screen**. The gRNA library-expressing A549 cells (hereinafter named A549-CRISPR cells) were constructed as described in Moffat's lab protocol. Briefly, $5 \times 10^7$ trypsinized A549 cells (calculated for 200-fold TKOv3 library coverage) were prepared in DMEM containing 10% FBS, supplemented with 8 μg mL$^{-1}$ polybrene to enhance lentiviral infection. Cells were incubated with lentiviral particles at MOI of 0.3. After 24 h of incubation, media were replaced with DMEM containing 10% FBS supplemented with 2 μg mL$^{-1}$ puromycin and cells were additionally incubated for 48 h to select transfected cells. The library was frozen at −80 °C before the screen.

For the screen, $15 \times 10^6$ A549-CRISPR cells were plated on a 15-cm dish and infected with PA14 for 90 min at MOI of 10 to reach 20–30% of surviving cells. To stop the infection, cells were trypsinized after washing and reseeded in DMEM supplemented with 10% FBS and 20 μg mL$^{-1}$ polymixin B. The surviving population was expanded and evaluated daily to monitor the recovery of cells. The day after the seeding, 90 μg mL$^{-1}$ gentamicin was also added in the medium to

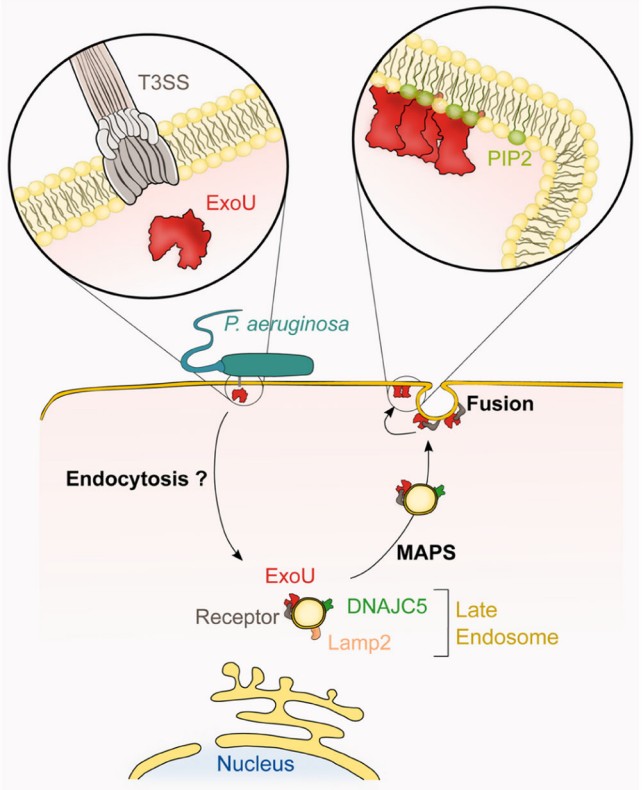

**Fig. 7 Proposed model of ExoU trafficking in host cells.** Upon delivery into the host cytoplasm by the T3SS, the toxin uses an endocytic pathway to reach the perinuclear region, as suggested by ExoU/EEA1 colabeling. Then, ExoU binds to the late endosome's limiting membrane (decorated by Lamp2 and DNAJC5), thanks to the interaction of ExoU with an as yet unidentified specific receptor at the vesicle's surface. ExoU remains at the external side of the vesicle's membrane and co-opts the DNAJC5-dependent MAPS pathway to achieve anterograde transport toward the cellular periphery (where vesicles loose Lamp2), and eventually the plasma membrane (PM). Fusion of vesicles with the PM brings ExoU close to PM's inner leaflet, where its membrane localization domain binds to PI(4,5)P2. PI(4,5)P2 binding triggers conformational changes in ExoU, leading to toxin oligomerization and activation of its phospholipase activity, which eventually induces PM rupture.

prevent proliferation of polymyxin-resistant bacteria. When surviving cells reached 70–80% confluence, they were subjected to a second and a third round of infection with PA14, allowing the repetition of the procedure with the same library coverage. The screen on A549-CRISPR cells was done in biological triplicates.

**Genomic DNA extraction, sequencing, and analysis**. After three rounds of infection, the surviving populations were expanded to obtain $2 \times 10^7$ cells. Uninfected A549-CRISPR cells were expanded similarly. Each cell population was subjected to genomic DNA (gDNA) extraction with a QIAamp DNA Blood Maxi kit (Qiagen, #51194) by following the manufacturer's protocol. Then, a one-step PCR was carried out to enrich and amplify the sequences corresponding to gRNAs from the genome of selected cell populations with Illumina TruSeq adapters (i5 and i7 indices, Supplementary Table 3). Moreover, a unique barcode of two 8-bp index

required for Illumina sequencing was added during the one-step PCR to each pool of amplified gRNAs. The PCR conditions for 50 μL were 25 μl of GoTaq G2 Hot Start Green Master Mix (#M7423, Promega), 0.5 μM of each primer (Supplementary Table 3) and 2.5 μg of gDNA. Three PCR reactions were performed simultaneously per sample to obtain sufficient quantities of amplified products. The PCR program was: denaturation at 95 °C for 2 min, followed by 26 cycles at 95 °C for 30 s, 55 °C for 30 s and 65 °C for 30 s, and a final elongation step at 65 °C for 5 min. After PCR amplification, each 50-μL reaction from the same sample were pooled, electrophoresed and the 200-bp bands were excised from agarose gel slice using Monarch DNA Gel extraction Kit (NEB, #T1020S). Then, each sequencing library was quantified on both NanoDrop and Qubit and a quality control of DNA was performed on Agilent Bioanalyzer system. Finally, a high throughput sequencing was performed on the pool of amplicons using a NextSeq 500 device (Illumina) at the CNRS platform of Orsay (Institut de Biologie Intégrative de la Cellule) and raw data were processed and analyzed using a web-based analysis platform named CRISPR-AnalyzeR. The adjusted p-values were analyzed according to MAGeCK method to identify overrepresented genes targeted by gRNAs in output compared to input.

**Cytotoxicity assay.** For infection assays, $1.5 \times 10^4$ A549 cells or derivatives were seeded per well in a 96-well plate 48 h before infection in DMEM supplemented with 10% FBS and 200 μg mL$^{-1}$ neomycin for transfectants. Thirty minutes before infection, Syto24 was added to the medium at 0.5 μM to label cell nuclei. Then, medium was removed and replaced by DMEM supplemented with PI at 1 μM. Cells were subsequently infected at MOI 20 with bacteria (OD1) unless indicated. The kinetics of PI incorporation was followed using an IncuCyte live-Cell microscope (Sartorius). Acquisitions were done every 15 min for 5 h using a ×10 objective. Images from bright field (phase), green channel (acquisition time 200 ms) and red channel (acquisition time 400 ms) were collected. The percentage of necrotic cells was calculated by dividing the number of PI-positive cells by the number of Syto24-positive cells.

For the cell retraction assay, cells were labeled with the CellTracker Red CMTPX (1 μM). Images were treated with ImageJ software. Briefly, images of CellTracker staining were binarized and total cell surface was calculated for 6 images at each time point.

**Purification of His6-ExoU.** The *exoU* gene amplified from the PP34 isolate (GESPA collection[62]) was cloned in pET15b. The obtained plasmid pET15b-*exoU* was introduced into *E. coli* BL21 (DE3) Star (InVitrogen). The induction was obtained by adding IPTG at 1 mM concentration during 3 h. The bacterial pellet, resuspended in 25 mM Tris-HCl, pH8, 500 mM NaCl and 10 mM imidazol, was disrupted using a Microfluidizer. Purification of His6-ExoU was performed on AKTA purifier using a HiTrap HP 5-ml column (GE HealthCare) with a step gradient of imidazole. Fractions, eluted with 200 mM imidazole and containing the ExoU protein, were loaded onto HiLoad SD200 16/60 prep column for a second step of purification.

**Antibodies and reagents.** The rabbit polyclonal antibody targeting human CSPα (DNAJC5) was purchased from ThermoFisher (#PA1-776, dilution 1:500). The mouse polyclonal antibodies against β-actin (#A1978, dilution 1:5,000), β-tubulin (#T0198, dilution 1:1,000) and FLAG (#F1804, dilutions: 1:5,000 for Western blot, 1:500 for immunolabeling) were purchased from Sigma-Aldrich. Specific antisera for ExoU were obtained in rabbits with 50 μg of purified full-length recombinant ExoU (kind gift from S. Barzu, Sanofi-Pasteur). The specific antibodies were affinity-purified on His6-ExoU (dilutions: 1:10,000 for Western blot, 1:500 for immunolabeling). The mouse monoclonal antibody targeting CSP in *Drosophila melanogaster* (named DCSP-1) was purchased from DSBH (#ab49, dilution 1:100). DCSP-1 (ab49) was deposited to the DSHB by Buchner, E./Hofbauer, A. The monoclonal mouse Hsc70 (HSPA8) and Hsp70 (HSPA1A) antibodies were purchased from R&D Systems (#MAB4148 and #MAB1663-SP, respectively, both diluted 1:5,000). The anti-Lamp2 antibody was from BD Transduction Laboratories (#555803, dilutions: 1:1,000 for Western blot, 1:500 for immunolabeling). Neomycin (#10131-035), Syto24 (#S7572) and propidium iodide (PI)(#P3566) were purchased from GIBCO, ThermoFisher and InVitrogen, respectively. The CellTracker Red CMTPX (#C34552) and CCF2-AM (K1032) were from ThermoFisher. Quercetin (#Q4951) and probenecid (#P8761) were from Sigma-Aldrich. siRNAs against Hsp70 (HSPA1, #s6968) and Hsc70 (HSPA8, #s6985) were purchased from ThermoFisher. The Complete inhibitor cocktail was from Roche (#04693159001) and the Micro-BCA Protein Assay Kit from ThermoFisher (#23235). The Lipofectamine RNAiMax kit was from Invitrogen (#13778-075).

**Eukaryotic cell lines and growth conditions.** The human embryonic kidney (HEK) 293 T (ATCC CRL-3216) and A549 (ATCC CCL-185) cells and their derivatives were cultured in Dulbecco's modified Eagle's medium (DMEM) supplemented with 10% fetal bovine serum (FBS). Cells were grown at 37 °C with 5% CO$_2$ and routinely passaged when reaching 70 to 80% confluence. Transfected cells with pLVX-*IRES-neo* vector, containing the wild-type or mutated *DNAJC5-FLAG* gene, were maintained by addition of 200 μg mL$^{-1}$ neomycin to the supplemented medium.

**Knockout of *DNAJC5* gene.** A pair of oligonucleotides corresponding to the enriched gRNA targeting *DNAJC5* gene (Supplementary Table 4) were annealed and cloned into the *BsmBI*-digested pLentiCRISPRv2 vector. Then, as previously described for TKOv3 library, lentiviral particles containing the recombinant plasmid pLentiCRISPRv2-gRNA-*DNAJC5* were created using the HEK293T cells and the psPAX2 and pMD2.G plasmids. Lentiviruses containing the pLentiCRISPRv2 vector without gRNA were also produced and referred to as empty vector (EV). A549 cells were then infected with these lentiviruses using 2 mL of lentiviral particles per 10-cm dish with cells at 50% confluence. Transfected cells were selected with 2 μg mL$^{-1}$ puromycin for 48 h and clones were isolated by limiting dilution. Clones were selected for their absence of DNAJC5 expression by Western blot and immunofluorescence. One clone was selected for further experiments.

**Complementation of deficient cells with *DNAJC5-FLAG* gene and its derivatives DNAJC5$^{H43Q}$ and DNAJC5$^{S10A-S34A}$.** The *DNAJC5* wild-type gene was synthetized by the Genewiz company and cloned into pUC57 plasmid. Some modifications have been introduced, without changing the amino acid sequence, to prevent the Cas9 endonuclease from cleaving the gene when inserted into the genome of deficient cells. Moreover, the 3X-FLAG tag was designed upstream of the gene and *EcoRI* and *BamHI* restriction sites were added upstream and downstream of the *DNAJC5-FLAG* gene respectively (Supplementary Table 5). The gene was inserted into pLVX-IRES-neo vector in *BamHI* and *EcoRI* sites. Lentiviral particles containing the empty vector pLVX or the pLVX-*DNAJC5-FLAG* plasmid were produced as previously described. Thereafter, $2.6 \times 10^6$ DNAJC5$^{-/-}$ cells were seeded per 10-cm dish the day before and were infected with 2 mL of lentivirus for 24 h. Then, media were replaced and transfected cells were cultured in DMEM containing 10% FBS and supplemented with 800 μg mL$^{-1}$ neomycin for at least 8 days before decreasing the antibiotic concentration to 200 μg mL$^{-1}$. In the same way, H43Q and S10A-S34A mutations on *DNAJC5-FLAG* gene were introduced by synthetizing the mutated genes (Supplementary Table 5) and by infecting the deficient cells with lentiviral particles containing the mutated genes cloned into the pLVX vector. In each case, clones were isolated by limiting dilution and tested for DNAJC5 expression by Western blot and immunofluorescence.

**Site-directed mutagenesis of *DNAJC5*.** The QuikChange Site-Directed Mutagenesis Kit (Agilent, #200523) was used to perform mutations on *DNAJC5-FLAG* gene. Briefly, oligonucleotide primers containing the desired mutations flanked by unmodified nucleotide sequence were synthetized as recommended by the manufacturer guidelines (Supplementary Table 6). For mutagenesis, 125 ng of each of the two complementary oligonucleotides were used in a reaction volume of 50 μL containing 30 ng of pUC57-*DNAJC5*-FLAG plasmid, 1 μL of dNTP mix and 5 μL of 10X reaction buffer. Then 1 μL of *PfuTurbo* DNA polymerase at 2.5 U.μL$^{-1}$ was added and the mixture was PCR amplified using the following cycling parameters: 95 °C for 30 s followed by 25 cycles at 95 °C for 30 s, 55 °C for 1 min and 68 °C for 3 min, and a final elongation step at 68 °C for 5 min. Thereafter, the nonmutated plasmid was digested by adding 1 μL of DpnI at 10 U.μL$^{-1}$ directly to amplification reaction followed by an incubation at 37 °C for 1 h. The remaining plasmid containing the desired mutation was transformed into Quick Change XL1-Blue Supercompetent cells. Mutants were checked by DNA sequencing. Then, as previously described, the mutated *DNAJC5* gene was digested and cloned into the pLVX-IRES-neo vector and deficient cells were complemented with the mutated gene. Clones were isolated by limiting dilution and selected as above.

**Quercetin assays.** For infection assays, A549 cells were seeded as previously described in 96-well plate, 48 h before. Quercetin was freshly prepared at 100 mM in DMSO. The day of infection, Syto24 was added to the medium at 0.5 μM 2 h before infection, the medium was removed and replaced by DMEM supplemented with PI and quercetin at the indicated concentrations. Cells were then infected with bacteria at OD 1 and at MOI 10. The kinetics of PI incorporation was followed as previously described using an automatized microscope IncuCyte.

***Drosophila* mutant engineering and infection assay.** To silence the *Csp* gene in *Drosophila melanogaster*, RNA interference was used. In brief, two different transgenic fly lines CSP-KD1 (Stock #34168 from the Vienna *Drosophila* Resource Center) and CSP-KD2 (Stock #33645 from the Bloomington *Drosophila* Stock Center) expressing a shRNA or a long dsRNA targeting *Csp*, respectively, under the control of GAL4-responsive elements were used. These flies were crossed with transgenic flies expressing the *Gal4* gene under the control of a temperature-inducible promoter (hs-Gal4, stock #2077 from the Bloomington *Drosophila* Stock Center) at 25 °C. Flies expressing a siRNA targeting the firefly *Luciferase* gene (Stock #31603 from the Bloomington *Drosophila* Stock Center) were also crossed with the heat shock-Gal4 line and used as control. The progeny was reared at 25 °C until the flies were 7 to 10-day old and the *Gal4* gene expression was induced by three heat shocks of 1 h at 37 °C, each performed during three consecutive days, in order to silence the *Csp* gene. Although one cannot formally rule out the possibility that a leaky expression of the hs-Gal4 driver occurs during development and results in developmental *csp* knockdown, the fact that the CSP-KD flies did not display any evidence of neurological disorders and early death (not shown), as observed in csp mutants[37], strongly argues against it.

Flies were then infected with a stationary phase culture of PP34 strain (or PP34Δ*exoU* strain), diluted at OD 1, by thoracic needle pricking, as previously described[38], and their survival rates were monitored. To confirm the knockdown of CSP, 20 flies per condition were homogenized in RIPA lysis buffer with a Precellys 24 (Bertin instruments) using CK14 tubes containing ceramic beads at 2800 *g* for four cycles of 30 s. Lysates were analyzed by Western blot using the anti-*Drosophila* CSP antibody.

**Bacterial load in *Drosophila***. Only alive flies were used. Each fly was homogenized at indicated time points with a pestle in a 0.5 mL microtube containing 100 μL LB. After centrifugation, the supernatant was assayed for CFU contents after serial dilutions and plating onto Pseudomonas Isolation Agar.

**Phospholipase activity assay of ExoU**. The phospholipase activity of ExoU was performed, as reported previously[13], using the Cayman Chemical cPLA$_2$ kit (#765021). Briefly, 5 μL of purified 6His-ExoU protein at 1 mg mL$^{-1}$ (65 pmols) were used per well of a 96-well plate containing 5 μL of Assay Buffer and 5 μL of cytosolic or membrane fractions (normalized by A$_{280}$) from DNAJC5$^{-/-}$ cells and DNAJC5$^{-/-}$::DNAJC5 cells. Then, reactions were initiated by adding 200 μL of substrate solution containing 1.5 mM arachidonyl thiophosphatidylcholonie and by shaking the plate for 30 s followed by an incubation for 1 h at room temperature. Absorbance was monitored at 24 h at 405 nm with an automated plate reader (Spark 10 M by TECAN) after addition of 10 μL of a solution containing 25 mM 5,5-dithiobis (2-dibitrobenzoic acid) (DTNB). The PLA$_2$ activity of ExoU was also monitored in wells without extracts called "Blank wells". Experiments were performed in triplicate. The PLA$_2$ activity of ExoU was calculated with the following formula where 10 is the extinction coefficient for DTNB.

$$\text{Substrate hydrolysis} = \frac{[\text{Mean of Abs(sample)} - \text{Mean of Abs(blank)}]}{10 \times 65.10^{-3}(\text{nmol of ExoU})}$$

**Lactate dehydrogenase release**. LDH release in the cell supernatant was measured using the Cytotoxicity Detection Kit from Roche Applied Science, following the recommended protocol. Briefly, cells were seeded at 2.5 × 10$^4$ in 96-well plates two days before, and infected in non-supplemented EBM2. At different post-infection times, 30 μL of supernatant were mixed with 100 μL of reaction mix and OD was read at 492 nm. OD values were subtracted with that of uninfected cells and Triton-solubilized cells were used to determine the total LDH present in the cell culture.

**ExoU injection assay**. A549 or DNAJC5$^{-/-}$ cells were seeded in 96-well plates 2 days before infection at 1.5 × 10$^4$ cells per well. Cells were infected at MOI of 10 for 4 h with the PP34Δ*exoU* strain producing ExoU$^{S142A}$-Bla toxin, as reported previously[32]. Then, cells were washed with PBS containing 2.5 mM probenecid, and incubated with freshly prepared CCF2-AM solution (2 μM) for 90 min in the dark at room temperature. The CCF2-AM cleavage by ExoU$^{S142A}$-Bla translocation was measured by comparing the emitted fluorescence at 447 nm (green, uncleaved) and 530 nm (blue, cleaved) upon excitation at 405 nm.

***DNAJC5* sequence analysis in DNAJC5$^{-/-}$ cells**. Genomic DNA was isolated from A549 and DNAJC5$^{-/-}$ cells (10$^7$ cells) using the QIAamp DNA Blood Maxi kit (Qiagen, #51194) and by following manufacturer's protocol. To amplify the *DNAJC5* gene, PCR reactions were carried out with 2.5 μg of gDNA in GoTaq G2 Hot Start Green Master Mix (Promega, #M7423) containing 0.5 μM of primers DNAJC5-Fw and DNAJC5-Rev (see Supplementary Table 2). PCR conditions were: 2 min at 95 °C, followed by 30 s at 95 °C, 30 s at 56 °C and 1 min at 72 °C (26x), and 5 min at 72 °C. The PCR products were Sanger-sequenced.

**Western blot analysis**. Cells were washed twice with cold PBS and lysed with lysis buffer containing 1% Triton X-100, 50 mM Tris-HCl pH 7.4, 150 mM NaCl, 1 mM EDTA, Roche protease inhibitor cocktail as well as vanadate (1 mM) and okadaic acid (50 nM). The lysates were centrifuged at 18,400 *g* for 10 min at 4 °C and protein concentration in the supernatant was measured with the Micro-BCA Protein Assay Kit. Supernatants were then denatured using Laemmli buffer containing β-mercaptoethanol at 95 °C for 5 min. For gel electrophoresis, proteins were run using 10% or 4–12% polyacrylamide gels (BioRad, #3450123). Proteins were then transferred onto a PVDF membrane, using the BioRad semi-dry transfer apparatus, and incubated for 1 h with 5% non-fat dairy milk followed by overnight incubation at 4 °C with primary antibodies. Membranes were then probed with secondary HRP-antibodies for 90 min at room temperature. After washings, signals were detected using the Immobilon Western blot Substrate and the ChemiDoc MP Imaging System.

**siRNA transfection**. A549 cells were seeded at 1.5 × 10$^5$ cells per well in a 6-well plate (for cell lysate preparation) or at 1.5 × 10$^4$ cells per well in a 96-well plate (for cytotoxicity assay) 24 h prior to transfection. Transfection was carried out with the 3 pmols siRNAs per well of 96-well plate or 30 pmols per well of 6-well plates,

using the Lipofectamine RNAiMax kit according to the manufacturer's protocol. After 24 h of incubation, the media were replaced by DMEM containing 10% FBS. Experiments were performed at 48 h post-transfection.

**Cellular fractionation**. Cellular fractionation was performed as previously described[29], with some modifications. Briefly, infected or uninfected cells (2.10$^7$) were washed with ice-cold PBS and scraped in 800 μL of buffer A (10 mM Tris-HCl pH 7.4, 10 mM KCl, 2 mM MgCl, 1 mM DTT, with protease inhibitor cocktail). The cellular suspension was supplemented to 250 mM sucrose to prevent sub-cellular organelle damage. Cells were fragmented by passing 20 times through a ball bearing homogenizer (8.020-mm bore, EMBL, Heidelberg, Germany) with an 8.006-mm ball bearing. The cellular homogenate was centrifuged at 1,000 × g for 10 min to remove nuclei and intact cells. The supernatant was then ultra-centrifuged at 100,000 × g for 30 min to sediment total microsome membranes. The pellet was resuspended in buffer A with sucrose, and was conserved at −20 °C, as well as the soluble fraction. Samples were normalized with OD$_{280}$.

**Immunofluorescence microscopy**. For ExoU, DNAJC5, FLAG and Lamp2 stainings, 5 × 10$^5$ A549 cells or derivatives were seeded in each well of a 24-well plate 48 h before fixation. For the co-staining with ExoU, cells were infected with CHA-*ExoU$^{S142A}$* or CHA-*ExoU$^{S142A-K178R}$* for 4 h at MOI of 10. Cells were fixed with 4% paraformaldehyde (PFA) for 15 min at room temperature and permeabilized with 0.5% Triton X-100 in PBS containing 4% PFA for 5 min. Cells were then stained using standard procedure with appropriate primary and secondary antibodies. Then, cells were counterstained with Hoechst. Images were collected on a Zeiss LSM880 confocal equipped with a Zeiss Plan-APO x63 numerical aperture 1.4 oil immersion objective. Successive planes in 3D stacks were taken every 0.2 μm. Images of z-sections were analyzed using the Intensity profile module from Icy software.

**Expression of DNAJC5-EGFP in HUVECs**. Human umbilical vein endothelial cells (HUVECs) were prepared in a previous study and frozen[65]. Cells were thawed and cultured in endothelial-basal medium 2 (EBM-2; Lonza) supplemented as recommended by the manufacturer. HUVECs were transfected using nucleofection (Amaxa, Lonza) according to the manufacturer's protocol. Briefly, the day before transfection, cells were seeded at a density of 30,000 cells/cm2 in EBM-2 medium supplemented with EGM-2 SingleQuots. Cells (2 × 10$^6$) were pelleted by centrifugation (6 min at 1,700 *g*) prior to being resuspended in 100 μL of Nucleofector® solution, mixed with 2.5 μg DNAJC5-EGFP plasmid[52]. The cells were nucleofected by using the program A-034. DNAJC5-EGFP-transfected cells were used at 24 h post-transfection.

**Statistics**. Statistics on genomic screening data are described above.

GraphPad 7.04 software was used for all other statistical analyses. For cytotoxicity studies or cell retraction assay, no statistical test was used. For Bla activity assay, a Student's *t* test was used between conditions using ExoU-Bla. For PLA2 activity test, a one-way ANOVA was employed, followed by Tukey's post-hoc test for data comparison. For the two latter assays, data distribution was normal according to Shapiros–Wilk's test and the tests were two-sided. For fly infection, a Log-Rank test was used. Data were considered significantly different when $p < 0.01$. For CFU comparison in PP34- vs PP34Δ*exoU*-infected flies, a Mann–Whitney's test was used. The number of experimental repetitions for each figure is shown in Supplementary Table 7.

**Reporting summary**. Further information on research design is available in the Nature Research Reporting Summary linked to this article.

## Data availability

The sequencing data generated in this study have been deposited in the NCBI Gene Expression Omnibus (GEO) and are accessible through GEO accession number GSE154751. Other source data are provided as a Source Data file. Source data are provided with this paper.

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

## Acknowledgements

The authors are grateful to Luke Chamberlain for the gift of the EGFP-DNAJC5 expression plasmids, to Simona Barzu for the ExoU antibody, Michel Ragno for ExoU purification. The authors thank Prof S. Lory (Microbiology Department, Harvard Medical School) for providing PA14 and Tn_exoU strains. This work, as well as V.D.'s PhD fellowship, were supported by a grant from the Fondation pour la Recherche Medicale (Team FRM 2017, DEQ20170336705). Confocal microscopy was performed at the μLife facility of the Interdisciplinary Research Institute of Grenoble (IRIG)/Department of Structural and Cellular Integrative Biology (DBSCI), funded by CEA Nanobio and Labex GRAL. Part of this project has been performed at the CMBA molecular screening platform and received funding from GRAL, a program from the Chemistry Biology Health (CBH) Graduate School of University Grenoble Alpes (ANR-17-EURE-0003). International Health Management Association (IHMA, USA) kindly provided the *P. aeruginosa* IHMA879472 strain. We thank Laurence Aubry and Agnès Journet for helpful discussions.

## Author contributions

V.D. and S.B. performed most experiments and analyzed data. P.H. and V.J. performed some experiments. E.T. and M-O.F. conceptualized the *Drosophila* experiments and managed the fly facility. I.A. created the bacterial tools. P.H. conceptualized the project and wrote the manuscript.

## Competing interests

The authors declare no competing interests.
