## [Peer Review File · Nature Communications]

REVIEWER COMMENTS

Reviewer #1 (Remarks to the Author):

Deruelle, V. et al.

This is a well-written article describing a genetic study that identified DNAJC5, a host chaperone, as a protein that is important for ExoU trafficking to the plasma membrane. DNAJC5 was shown to play a role in toxicity by several independent experiments that include the CRISPR screen for toxin-resistant cells, a specifically targeted knock out, an engineered strain of *Drosophila* for an in vivo study and the use of chemical inhibitor. DNAJC5 does not seem to impact intoxication by ExoS and hence is not interfering with T3SS translocation, nor does it impact intoxication by the ExlA toxin, a pore-forming protein. DNAJC5 is not required for membrane or substrate binding. Other than trafficking, DNAJC5 may be contributing to transient intracellular stability of ExoU, perhaps as demonstrated by the increased toxicity of the DNAJC5 H43Q allele (gain of function allele). The finding of a host chaperone that shuttles the toxin to the plasma membrane is novel and could represent an important tool to study both DNAJC5 and bacterial effectors injected by the T3SS. Overall, the comments are minor but the manuscript could be improved by addressing some points about enzymatic activity, including prior trafficking data in the discussion and clarifying text for the following comments:

1. The authors suggest that host-cell mechanisms protect from self-toxicity. This appears to be a special situation where the toxin has cross-kingdom substrate specificity, which usually occurs in T6SS systems where there is an immunity protein. Referencing some of these initial statements might make the point clearer.
2. Line 65, the addition of
3. Figure 1b. Comparison of infected and uninfected cells is spelled out in the legend but neither panel a nor panel b seems to specifically illustrate/report this comparison.
4. Figure 1f. PP34 seems to have a much higher background toxicity in the vector control transfectants. This result suggests that the amount of ExoU being delivered might be able to eventually overcome DNAJC5. Is there evidence that DNAJC5 inhibits ExoU phospholipase activity in vivo? This hypothesis would also fit with the Quercetin titration data.
5. Several sentences seem incomplete, e.g. Line 166-167.
6. Since DNAJC5 H43Q significantly enhances PI staining (Figure 4), is it possible that transfecting some DNAJC5 alleles alone is toxic (without bacterial infection or bacteria without an effector)? Were these controls done?
7. Line 192-193. It is unclear whether the authors have evidence to conclude the Hsc70/Hsp70 delays ExoU-dependent cell lysis. Please clarify.
7. Supplementary Figure 5. Poor expression by the L115R and Δ L116 may account for some of the changes in protein trafficking or toxicity.
8. Abstract lines 23-24, Line 238 and lines 297-301. The authors present no direct evidence for the activation of ExoU by PI(4,5)P2 making these comments unsupported by evidence presented in their manuscript. PI(4,5)P2 has been documented to enhance activity, but it is also been shown that, ExoU can target prokaryotic membranes as well as liposomes devoid of PI(4,5)P2 making PI(4,5)P2 not required for activation. Importantly, the preparations used for measuring enzymatic activity in this study and others are significantly contaminated with many proteins and lipids. Finally, product accumulation for 24 h seems like a prolonged time period for an activity assessment. In other words, the commercially available enzymatic assay appears to be measuring a minor activity under suboptimal conditions. Some of the observations regarding activation (no activity with a soluble extract) could be due to a suboptimal enzyme assay. Overall, careful

interpretation of enzyme activity data, the development and optimization of an assay and balancing activation with other biochemical activities, including this novel trafficking pathway will likely lead to new biological information for both ExoU-like toxins and DNAJC5-like proteins.

8. Previous studies on ExoU trafficking indicated that the protein was associated with acidic organelles and the early endosome marker EEA1. EEA1 association was dependent on K178-ubiquitin modification. Since there are two populations of molecules is it possible that the authors are assaying unmodified ExoU? How do these prior studies build upon the trafficking story presented in this manuscript? Not much is discussed except co-localization with LAMP2.

9. The title indicates that necrosis is being measured when only PI incorporation is used. The authors should also perform tests for release of LDH to confirm necrotic cell death and actual cellular lysis.

Reviewer #2 (Remarks to the Author):

The manuscript by Deruelle et al reports the identification of DNAJC5 (also named CSP) as an essential mediator for bacterial toxin ExoU-induced necrosis. The authors use a genome-wide CRISPR screen to search for genes when inactivated could protect cells from ExoU-induced cell death. The screen resulted in only one confident hit, which is the heat shock protein 70 (Hsp70) co-chaperone DNAJC5. DNAJC5 was previously shown to regulate exocytosis and an unconventional protein secretion pathway named misfolded-associated protein secretion. It was localized to the membrane of late endosome/lysosome in non-neuronal cells or in synaptic vesicles in neurons. The authors show that ExoU co-localizes with DNAJC5 on vesicle membranes and that DNAJC5 appears to be required for the trafficking of the toxin to the plasma membrane. This trafficking process was abolished by DNAJC5 mutations that have been linked to ceroid lipofuscinosis, a neurological disorder. The authors propose that DNAJC5 escorts ExoU from endolysosomes to the plasma membrane, which is required for its toxicity. Overall, this is an interesting study reporting a new trafficking itinerary for a bacterial toxin that is critically relevant to nosocomial infections. The data presented are largely convincing. Most conclusions are justified with a few exceptions (see below). The following concerns should be addressed either by additional experiments or clarification.

Specific points:

1. The authors show that ExoU is partially co-localized with DNAJC5 at late endosomes. However, they could not detect any interactions between these proteins. Moreover, the membrane localization of ExoU remains unaffected in DNAJC5 knockout cells. For these reasons, the conclusion that DNAJC5 escorts ExoU to the plasma membrane (line 209) is not justified.
2. In fact, the authors never show that ExoU is present in the plasma membrane in wild-type cells infected with ExoU. The conclusion that DNAJC5 is required for the plasma membrane delivery of ExoU is based on indirect evidence (DNAJC5 mutants defective in cell surface trafficking only partially rescue ExoU-induced toxicity). Can the authors use TIRF microscopy to show the kinetics of this trafficking process and that in DNAJC5 mutant cells, this process is compromised? Without these results, the main conclusion of the paper is quite weak.
3. Line 171, "in these cells, both DNAJC5-GFP and ExoU were localized at the vesicles limiting membrane and were not intraluminal". The resolution of the imaging data is not sufficient to make this conclusion. If one examines the data carefully, there seems to be some red and green dots present in the lumen of these vesicles in Fig. 3d.
4. Figure 3c, can the authors comment on why the DNAJC5-localized vesicles appear elongated in these cells? Additionally, although the authors analyzed the localization of DNAJC5 re-expressed in DNAJC5 knockout cells, the protein is still overexpressed at a much higher level compared to endogenous DNAJC5. Thus, the unusual vesicle structure as well as the partial co-localization with LAMP2 may be an artifact of this overexpression approach. Ideally, the authors should analyze endogenous DNAJC5 with either a specific antibody or by CRISPR-mediated endogenous tagging.
5. Page 10, the authors examined the localization of two DNAJC5 disease mutants. They claimed that these mutants are NOT localized to endolysosomal vesicles. However, judging from Fig. S4, there seem to be some degree of co-localization with LAMP2.
6. Figure 5, the rescue activity of WT DNAJC5 and various mutants should be normalized by the

expression levels of DNAJC5 variants. Additionally, the authors mentioned that the DNAJC5 delJ mutant failed to completely rescue toxin-induced cell death. The J domain is a highly conserved HSC70 interacting domain. Please comment on why the H43Q mutant is as active as wild-type DNAJC5, but the delta J mutant is inactive, although both mutations disrupt HSC70 binding.

7. Fig. S3b, please quantify the knockdown efficiency. As *Drosophila* has three CSP isoforms, please comment on whether the siRNA constructs target all three isoforms or not.
8. Figure 2, the labels are confusing. The experiments also miss uninfected control KD flies.
9. Line 102, the assay used a fluorogenic substrate, which should not be mixed with a "FRET substrate".
10. In supplementary Fig4, the label for "L115A" should be "L115R"
11. Typos and grammar errors: line 166, the sentence "However," reads like an incomplete sentence. Line 101, "used" should be deleted. Figure 1 legend, c, should be "6-replicates", not "6-eplicates".
12. Please provide a table listing all the essential reagents including antibodies, chemicals and recombinant DNA. Please also indicate the number of experimental repeats in the figure legends or in a separate session per the journal policy.

Reviewer #3 (Remarks to the Author):

In this work, the authors have performed a CRISPR-Cas9 screen on cultured cells to identify genes required for *P. aeruginosa* ExoU-induced necrosis and identified a single gene encoding DNAJC5/CSPalpha, known to be involved in an unconventional secretory pathway known as MAPS. They show that this host gene is required for bacterial virulence in cultured cells and in vivo using a *Drosophila* systemic infection model. They then perform immunohistochemistry as well as some biochemistry on cultured cells to show that ExoU distribution overlaps that of DNAJ5 on the surface of intracellular vesicles, although the host protein is not required for the membrane-associated distribution of ExoU. They next elegantly demonstrate that the co-chaperone activity of DNAJC5 is not required for ExoU toxicity. Instead, it appears that the trafficking activity of DNAJC5 is important for ExoU toxin activity, even though experiments aimed at determining whether there is a direct interaction between the toxin and the co-chaperone were not conclusive.

This study is interesting as it uncovers a novel host factor required to mediate the toxicity of the exoU virulence factor.

Since the PP34 strain is highly virulent and leads to some lysis of DNAJC5 mutant cells, it would be important to perform a control experiment using an exoU-mutant PP34 to show that this lysis is indeed exoU-independent in DNAJC5 mutant cells. The same experiment should be performed in *Drosophila* and it would be expected that the wt and exoU *P. aeruginosa* would kill CSP mutants at the same rate. (With respect to Fig. 2, it would be desirable to display the whole survival curves until all infected flies are killed).

As the authors use a hsp driver to inactivate Csp, there is a concern that the effect might be indirectly caused by developmental effects caused by partial gene inactivation, as the hsp promoter is leaky (in addition, the temperature at which the crosses were made is not indicated). It would be needed to exclude this possibility using either a Gene Switch ubiquitous driver or a Gal4-Gal80ts system.

The rate of fly demise upon PP34 challenge is impressive (Fig. 2b), yet somewhat variable (LT50 oscillating between 3-4 hours to 8 hours). It is not clear whether PP34 bacteria would have the time to grow significantly in this short interval of 3 hours. It would be thus needed to also use the PA14 strain that does not kill as fast (and is well-characterized in the *Drosophila* model) and monitor the bacterial burden at a couple of time points, when at least 80% of the wt and mutant flies are still alive to check that there is no difference of growth between the wt and exoU strain in

vivo.

The pathology induced by *P. aeruginosa* in *Drosophila* is not well documented. It would be interesting to inject propidium iodide (or SYTOX green) along with PP34 to assess whether there is necrosis occurring in fly tissues, which should be decreased in the *exoU*-infected flies.

Minor points

Fig. 3c, 3rd inset: is the profile really corresponding to the segment shown in the inset? One does not really see where the structure corresponding to the first peaks (green and red) is located on the picture. It would generally be helpful to orient the segments with an arrow.

Fig. S1: It would be nice to display also the protein sequence and show where the premature STOP codon lies with respect to the protein.

Typo at lines 65 (adding)

Gentamicin not gentamycin.

The number of times experiments have been performed should be stated, especially survival experiments.

Response to the Reviewers' comments

First of all, I would like to thank the reviewers to have accepted to evaluate our manuscript and to have taken the time to do it thoroughly. We were glad to read the general positive comments on this work.

We performed new experiments, as suggested, and added several control experiments. Therefore, there are new supplementary figures for the control of toxicity experiments and for the role of ExoU ubiquitylation (Supplementary Figs. 2 and 6, respectively). Some previous figures have been completed.

Reviewer #1 (Remarks to the Author):

Deruelle, V. et al.

This is a well-written article describing a genetic study that identified DNAJC5, a host chaperone, as a protein that is important for ExoU trafficking to the plasma membrane. DNAJC5 was shown to play a role in toxicity by several independent experiments that include the CRISPR screen for toxin-resistant cells, a specifically targeted knock out, an engineered strain of Drosophila for an in vivo study and the use of chemical inhibitor. DNAJC5 does not seem to impact intoxication by ExoS and hence is not interfering with T3SS translocation, nor does it impact intoxication by the ExlA toxin, a pore-forming protein. DNAJC5 is not required for membrane or substrate binding. Other than trafficking, DNAJC5 may be contributing to transient intracellular stability of ExoU, perhaps as demonstrated by the increased toxicity of the DNAJC5 H43Q allele (gain of function allele). The finding of a host chaperone that shuttles the toxin to the plasma membrane is novel and could represent an important tool to study both DNAJC5 and bacterial effectors injected by the T3SS. Overall, the comments are minor but the manuscript could be improved by addressing some points about enzymatic activity, including prior trafficking data in the discussion and clarifying text for the following comments:

1. The authors suggest that host-cell mechanisms protect from self-toxicity. This appears to be a special situation where the toxin has cross-kingdom substrate specificity, which usually occurs in T6SS systems where there is an immunity protein. Referencing some of these initial statements might make the point clearer.

This is just a general comment, at the beginning of the Introduction, on the fact that bacterial virulence factors are not self toxic. One of the mechanisms preventing self toxicity is the requirement of a host factor or several, as for ExoU, that targets toxicity to the host cell. We believe that discussing further this notion would be out of the scope of the Introduction.

2. Line 65, the addition of

This has been modified.

3. Figure 1b. Comparison of infected and uninfected cells is spelled out in the legend but neither panel a nor panel b seems to specifically illustrate/report this comparison.

We added an arrow in Fig. 1a to clarify this point.

4. *Figure 1f. PP34 seems to have a much higher background toxicity in the vector control transfectants. This result suggests that the amount of ExoU being delivered might be able to eventually overcome DNAJC5. Is there evidence that DNAJC5 inhibits ExoU phospholipase activity in vivo? This hypothesis would also fit with the Quercetin titration data.*

There are two arguments against a putative inhibitory activity of DNAJC5 on ExoU: 1/ DNAJC5 and ExoU do not interact and 2/ the presence of DNAJC5 does not alter ExoU phospholipase activity in vitro (Fig. 6). However, we cannot completely rule out this hypothesis in cellulose.

5. *Several sentences seem incomplete, e.g. Line 166-167.*

The sentences have been completed.

6. *Since DNAJC5 H43Q significantly enhances PI staining (Figure 4), is it possible that transfecting some DNAJC5 alleles alone is toxic (without bacterial infection or bacteria without an effector)? Were these controls done?*

This is indeed a good point. These controls had not been done and they are now provided in Supplementary Fig. 2a,b using ExoU isogenic mutants. PA14 Δ exoU did not induce necrosis in DNAJC5^{H43Q}, while PP34 Δ exoU did (Supplementary Fig. 4a,b); no PI incorporation was detected when cells were incubated without bacteria (not shown). Thus, the H43Q mutation seems to increase the sensitivity of cells to *P. aeruginosa*'s virulence factors other than ExoU. This secondary toxicity likely explains the increased necrotizing activity of ExoU in DNAJC5^{-/-}::DNAJC5^{H43Q} cells (Fig. 4b).

7. *Line 192-193. It is unclear whether the authors have evidence to conclude the Hsc70/Hsp70 delays ExoU-dependent cell lysis. Please clarify.*

This part of the sentence was cancelled according to above results.

7. *Supplementary Figure 5. Poor expression by the L115R and Δ L116 may account for some of the changes in protein trafficking or toxicity.*

Although the images shown in Fig. 5 seem to indicate lower protein amounts, expression levels of the two mutants L115R and Δ L116 are comparable to that of the complemented wild type in Western blot (Supplementary Fig. 1d), except that they form aggregates. We rather think that these mutated forms are more concentrated in late endosomes.

8. *Abstract lines 23-24, Line 238 and lines 297-301. The authors present no direct evidence for the activation of ExoU by PI(4,5)P2 making these comments unsupported by evidence presented in their manuscript. PI(4,5)P2 has been documented to enhance activity, but it is also been shown that, ExoU can target prokaryotic membranes as well as liposomes devoid of PI(4,5)P2 making PI(4,5)P2 not required for activation.*

The reference to PI(4,5)P2 has been removed from the abstract, as it is not a result of our work. There are compelling evidence in the literature showing that PI(4,5)P2 can bind and activate ExoU (references in the manuscript), which does not exclude that other molecules (lipids) can also activate ExoU. Therefore, according to the reviewer's comment, we modified the text in the two other locations indicated to let this possibility open.

Importantly, the preparations used for measuring enzymatic activity in this study and others are significantly contaminated with many proteins and lipids. Finally, product accumulation for 24 h

seems like a prolonged time period for an activity assessment. In other words, the commercially available enzymatic assay appears to be measuring a minor activity under suboptimal conditions. Some of the observations regarding activation (no activity with a soluble extract) could be due to a suboptimal enzyme assay. Overall, careful interpretation of enzyme activity data, the development and optimization of an assay and balancing activation with other biochemical activities, including this novel trafficking pathway will likely lead to new biological information for both ExoU-like toxins and DNAJC5-like proteins.

We used an assay that has previously been set up for ExoU by Hauser's lab (Tyson et al. J Biol Chem 290:2919, 2015), with the same incubation time. It is not as efficient as that reported by Dara Frank, and probably not as informative, notably to observe activation by the soluble fractions. However, in Fig. 6, we mostly wanted to compare ExoU activation using WT or DNAJC5-deficient membranes. In that respect, the assay was fully relevant.

8. Previous studies on ExoU trafficking indicated that the protein was associated with acidic organelles and the early endosome marker EEA1. EEA1 association was dependent on K178-ubiquitin modification. Since there are two populations of molecules is it possible that the authors are assaying unmodified ExoU? How do these prior studies build upon the trafficking story presented in this manuscript? Not much is discussed except co-localization with LAMP2.

Ubiquitylated ExoU on K178 was shown to colocalize preferentially with EEA1-positive endosomes (Gendrin et al Plos Path 2012) and the K178R mutation, preventing ubiquitylation, had no impact on ExoU toxicity (Stirling et al. Cell Mic 2006).

We had tested the effect of K178R mutation on ExoU subcellular localization and these data are now reported as Supplementary Fig. 6. We did not observe any alteration in ExoU colocalization with DNAJC5. Thus, this result, together with the fact that the K178R mutation does not impact toxicity, indicate that ExoU covalent ubiquitylation is not important for ExoU trafficking toward the plasma membrane.

9. The title indicates that necrosis is being measured when only PI incorporation is used. The authors should also perform tests for release of LDH to confirm necrotic cell death and actual cellular lysis.

LDH data are now provided in Supplementary Fig. 2c,d. These results confirm PI incorporation data.

Reviewer #2 (Remarks to the Author):

The manuscript by Deruelle et al reports the identification of DNAJC5 (also named CSP) as an essential mediator for bacterial toxin ExoU-induced necrosis. The authors use a genome-wide CRISPR screen to search for genes when inactivated could protect cells from ExoU-induced cell death. The screen resulted in only one confident hit, which is the heat shock protein 70 (Hsp70) co-chaperone DNAJC5. DNAJC5 was previously shown to regulate exocytosis and an unconventional protein secretion pathway named misfolded-associated protein secretion. It was localized to the membrane of late endosome/lysosome in non-neuronal cells or in synaptic vesicles in neurons. The authors show that ExoU co-localizes with DNAJC5 on vesicle membranes and that DNAJC5 appears to be required for the trafficking of the toxin to the plasma membrane. This trafficking process was abolished by DNAJC5 mutations that have been linked to ceroid lipofuscinosis, a neurological disorder. The authors propose that DNAJC5 escorts ExoU from endolysosomes to the plasma membrane, which is required for its toxicity.

Overall, this is an interesting study reporting a new trafficking itinerary for a bacterial toxin that is

critically relevant to nosocomial infections. The data presented are largely convincing. Most conclusions are justified with a few exceptions (see below). The following concerns should be addressed either by additional experiments or clarification.

Specific points:

1. The authors show that ExoU is partially co-localized with DNAJC5 at late endosomes. However, they could not detect any interactions between these proteins. Moreover, the membrane localization of ExoU remains unaffected in DNAJC5 knockout cells. For these reasons, the conclusion that DNAJC5 escorts ExoU to the plasma membrane (line 209) is not justified.

This is correct. ExoU is not escorted by DNAJC5 itself, but by DNAJC5-positive vesicles. The paragraph title has been modified accordingly.

2. In fact, the authors never show that ExoU is present in the plasma membrane in wild-type cells infected with ExoU. The conclusion that DNAJC5 is required for the plasma membrane delivery of ExoU is based on indirect evidence (DNAJC5 mutants defective in cell surface trafficking only partially rescue ExoU-induced toxicity). Can the authors use TIRF microscopy to show the kinetics of this trafficking process and that in DNAJC5 mutant cells, this process is compromised? Without these results, the main conclusion of the paper is quite weak.

We performed TIRF experiments on infected cells as suggested, and visualized the basal membrane of infected cells. However, bacteria rapidly moved under the cells and the bright ExoU immunofluorescence signals of the bacteria polluted any potential signals at the cell's membrane (Fig. 1 of this document). Therefore, we could not see any difference in labeling between A549 and DNAJC5^{-/-} cells with this technique. As far as we know, no publication reported the localization of ExoU at the plasma membrane by TIRF microscopy, which can only visualize the basal membrane. As ExoU targets the inner leaflet of the plasma membrane, it is not accessible for direct labeling. We don't know any other approach that we could use to precisely determine the amount of ExoU at the plasma membrane. Thus, we moderated our conclusions on the targeting at the plasma membrane in several locations of the manuscript. Nevertheless, it is well established that cargoes of DNAJC5-positive vesicles are not delivered in the extracellular milieu when DNAJC5 is mutated on L115/L116 and the mechanism of DNAJC5-dependent vesicle transportation has been partially characterized. Therefore, we believe that our results with the mutated versions of DNAJC5 strongly suggest that ExoU is transported to the plasma membrane by these vesicles. Some additional mechanisms of ExoU activation may occur at the vesicles' membrane, but we have no clue for this latter hypothesis.

Fig. 1. TIRF micrographs of A549 and DNAJC5^{-/-} cells infected with CHA-ExoU^{S142A} for 2 hours. Cells were labeled with ExoU antibody and phalloidin. ExoU signals were confined to bacteria located below the cells.

3. Line 171, “in these cells, both DNAJC5-GFP and ExoU were localized at the vesicles limiting membrane and were not intraluminal”. The resolution of the imaging data is not sufficient to make this conclusion. If one examines the data carefully, there seems to be some red and green dots present in the lumen of these vesicles in Fig. 3d.

We agree that the resolution is low (although using the 100X objective), but when we played with the focus under the microscope, we clearly observed that ExoU labeling was not intraluminal. We could also observe a membrane labeling in A549 cells when the vesicles were enlarged owing to L115R or Δ L116 mutations of DNAJC5 (Supplementary Fig. 5). However, we cannot conclude on the precise membrane localization: inner or outer leaflet.

4. Figure 3c, can the authors comment on why the DNAJC5-localized vesicles appear elongated in these cells? Additionally, although the authors analyzed the localization of DNAJC5 re-expressed in DNAJC5 knockout cells, the protein is still overexpressed at a much higher level compared to endogenous DNAJC5. Thus, the unusual vesicle structure as well as the partial co-localization with LAMP2 may be an artifact of this overexpression approach. Ideally, the authors should analyze endogenous DNAJC5 with either a specific antibody or by CRISPR-mediated endogenous tagging.

We do not know why these peripheral vesicles are elongated. They were also observed in PC12 cells (Greaves et al, J Biol Chem 2012). Unfortunately, DNAJC5 expression is too low in native cells to be observed by immunofluorescence technique. As these elongated vesicles are also observed in DNAJC5^{-/-} vesicles labeled with ExoU (Fig. 3b), we believe that these vesicles are likely not an artifact. It is possible that the geometry of these vesicles is modified, after the loss of Lamp2, during their migration along the intracellular trafficking pathway.

5. Page 10, the authors examined the localization of two DNAJC5 disease mutants. They claimed that these mutants are NOT localized to endolysosomal vesicles. However, judging from Fig. S4, there seem to be some degree of co-localization with LAMP2.

We reexamined the slides and there is indeed a partial colocalization. This is now indicated in the text. However, there is a striking difference with wild-type DNAJC5, which perfectly matches the Lamp2 labelling in the perinuclear area.

6. Figure 5, the rescue activity of WT DNAJC5 and various mutants should be normalized by the expression levels of DNAJC5 variants. Additionally, the authors mentioned that the DNAJC5 delJ mutant failed to completely rescue toxin-induced cell death. The J domain is a highly conserved HSC70 interacting domain. Please comment on why the H43Q mutant is as active as wild-type DNAJC5, but the delta J mutant is inactive, although both mutations disrupt HSC70 binding.

ExoU yielded similar intoxication curves in native A549 and DNAJC5^{-/-}::DNAJC5 cells (Fig. 1e,f), although DNAJC5 expression levels is higher in complemented cells. We also obtained similar curves with complemented cells expressing various levels of DNAJC5 (not shown). We interpreted these data as a plateau of ExoU toxic activity that is reached at low levels of DNAJC5 expression. Therefore, normalization of toxicity with DNAJC5 expression levels would not be correct.

The HPD motif is not essential for exocytosis, while it is required for the interaction with and activation of Hsp proteins (reviewed in Gundersen, Prog Neurobiol, 2020, 188:101758). Our results suggest that the J domain may also have a role in exocytosis, independent of the HPD motif. In our investigations, we screened different DNAJC5 mutants, searching for trafficking defects. This was the

only mutant to perturb vesicle trafficking, similar to what was observed with the two pathological mutations.

7. *Fig. S3b, please quantify the knockdown efficiency. As Drosophila has three CSP isoforms, please comment on whether the siRNA constructs target all three isoforms or not.*

The CSP/actin signals on Western blots have been quantified and shown in Supplementary Fig. 4b. There is a net decrease of CSP signals of CSP-KD1 and CSP-KD2 compared to the Gal4 control (0.44 and 0.38, respectively), however the wild-type controls in each experiments exhibited lower and variable CSP/actin signals (0.62 and 0.95, respectively), suggesting that genetic manipulation may influence CSP levels.

There is indeed only one *CSP* gene in *Drosophila*, but three isoforms; two of them having a large deletion in the middle of the mRNA sequence. The two RNAi are located outside of the large deleted sequence, so they can target all three isoforms.

8. *Figure 2, the labels are confusing. The experiments also miss uninfected control KD flies.*

The labels were modified on the figure. The uninfected controls correspond to mock-infected CSP-KD1 and CSP-KD2 flies with PBS. The infected controls are Luc-KD flies.

9. *Line 102, the assay used a fluorogenic substrate, which should not be mixed with a "FRET substrate".*

This has been corrected.

10. *In supplementary Fig4, the label for "L115A" should be "L115R"*

This has been corrected.

11. *Typos and grammar errors: line 166, the sentence "However," reads like an incomplete sentence. Line 101, "used" should be deleted. Figure 1 legend, c, should be "6-replicates", not "6-eplicates".*

Typos have been corrected.

12. *Please provide a table listing all the essential reagents including antibodies, chemicals and recombinant DNA. Please also indicate the number of experimental repeats in the figure legends or in a separate session per the journal policy.*

The reagent table is not mandatory for Nat Comm journal, which accepts a standard description of these molecules in the paragraphs. A Table showing the number of experimental repeats is provided in the revised version (Supplementary Table. 7).

Reviewer #3 (Remarks to the Author):

*In this work, the authors have performed a CRISPR-Cas9 screen on cultured cells to identify genes required for *P. aeruginosa* ExoU-induced necrosis and identified a single gene encoding DNAJC5/CSP α , known to be involved in an unconventional secretory pathway known as MAPS.*

They show that this host gene is required for bacterial virulence in cultured cells and in vivo using a Drosophila systemic infection model. They then perform immunohistochemistry as well as some biochemistry on cultured cells to show that ExoU distribution overlaps that of DNAJ5 on the surface of intracellular vesicles, although the host protein is not required for the membrane-associated distribution of ExoU. They next elegantly demonstrate that the co-chaperone activity of DNAJ5 is not required for ExoU toxicity. Instead, it appears that the trafficking activity of DNAJ5 is important for ExoU toxin activity, even though experiments aimed at determining whether there is a direct interaction between the toxin and the co-chaperone were not conclusive.

This study is interesting as it uncovers a novel host factor required to mediate the toxicity of the exoU virulence factor.

Since the PP34 strain is highly virulent and leads to some lysis of DNAJ5 mutant cells, it would be important to perform a control experiment using an exoU-mutant PP34 to show that this lysis is indeed exoU-independent in DNAJ5 mutant cells. The same experiment should be performed in Drosophila and it would be expected that the wt and exoU P. aeruginosa would kill CSP mutants at the same rate. (With respect to Fig. 2, it would be desirable to display the whole survival curves until all infected flies are killed).

The control experiment investigating the necrotizing activity of PP34 Δ exoU in DNAJ5^{-/-} cells (and all other complemented cell lines) is now shown in Supplementary Fig. 2b. PP34 Δ exoU is not toxic in DNAJ5^{-/-} cells, which indicates that the partial toxic activity observed in PP34-infected DNAJ5^{-/-} cells is due to ExoU. This is now commented in the text. Whether this residual toxicity is caused by ExoU alone or by synergy with a factor specific to PP34 strain was not examined.

Data showing the survival curves of *Drosophila* knocked-down for CSP and infected with PP34 Δ exoU is now shown in Supplementary Fig. 4c. Both PP34 and PP34 Δ exoU were poorly toxic in CSP-deficient flies (Fig. 2 and Supplementary Fig. 4c).

It was not possible to follow fly death at night (due to access restriction on the worksite). Therefore, survival curves could only be established for 16 hours maximum and gave significant results.

As the authors use a hsp driver to inactivate Csp, there is a concern that the effect might be indirectly caused by developmental effects caused by partial gene inactivation, as the hsp promoter is leaky (in addition, the temperature at which the crosses were made is not indicated). It would be needed to exclude this possibility using either a Gene Switch ubiquitous driver or a Gal4-Gal80ts system.

This is why we used mock-infected flies (with PBS injection) that control for both the CSP deficiency effect and needle injury. All crosses were performed at 25°C (indicated in the revised version).

The rate of fly demise upon PP34 challenge is impressive (Fig. 2b), yet somewhat variable (LT50 oscillating between 3-4 hours to 8 hours). It is not clear whether PP34 bacteria would have the time to grow significantly in this short interval of 3 hours. It would be thus needed to also use the PA14 strain that does not kill as fast (and is well-characterized in the Drosophila model) and monitor the bacterial burden at a couple of time points, when at least 80% of the wt and mutant flies are still alive to check that there is no difference of growth between the wt and exoU strain in vivo.

PP34 strain is indeed very toxic in *Drosophila* (and in cellular experiments), more than PA14. We used PP34 because there is strong difference in fly survival between infections with PP34 and PP34 Δ exoU (Supplementary Fig. 4c). Following the suggestion of the reviewer, we assayed PA14 vs PA14 Δ exoU

toxicity on flies. However, in our hands, the survival curves of flies infected with PA14 and PA14 Δ exoU were not significantly different, even at low OD values (Fig. 2 of this document), although PA14 Δ exoU is not toxic in A549 cells (Supplementary Fig. 2a). This result suggests that PA14 lethal effect in *Drosophila* is mostly mediated by virulence factors other than ExoU. Therefore, PP34 is more adapted to observe ExoU-dependent effects in this in-vivo model.

Fig. 2. Survival curves of flies infected by PA14, PA14 Δ exoU or mock-infected with PBS

There is indeed some variations in the LT50 from one experiment to another, a phenomenon which has been observed by us and others and may result from slight differences in bacterial OD and/or fly shape; this is why we include all controls in each experiment.

The bacterial burden at 2 and 6 hpi is now reported in Supplementary Fig. 4d. The CFU numbers were not significantly different between flies infected with PP34 and PP34 Δ exoU for the three backgrounds (Luc-KD, CSP-KD1 and CSP-KD2) at the two time points, except for CSP-KD1 flies at 2 hpi, pointing to a major role of ExoU in fly death, rather than a difference in bacterial growth.

The pathology induced by P. aeruginosa in Drosophila is not well documented. It would be interesting to inject propidium iodide (or SYTOX green) along with PP34 to assess whether there is necrosis occurring in fly tissues, which should be decreased in the exoU-infected flies.

Although it is an interesting point, we think it is out of the scope of this study.

Minor points

Fig. 3c, 3rd inset: is the profile really corresponding to the segment shown in the inset? One does not really see where the structure corresponding to the first peaks (green and red) is located on the picture. It would generally be helpful to orient the segments with an arrow.

We performed again the analysis to double-check all the profiles presented in the different figures. The 3rd profile of Fig. 3c was accurate, but the 4th profile was not correct and we modified it. In this

new analysis, ExoU and DNAJC5 profiles at the bottom match better and the text was modified. The segments were oriented with arrows.

Fig. S1: It would be nice to display also the protein sequence and show where the premature STOP codon lies with respect to the protein.

This information is now shown in Supplementary Fig. 1a.

*Typo at lines 65 (adding)
Gentamicin not gentamycin.*

This has been corrected

The number of times experiments have been performed should be stated, especially survival experiments.

A table is now provided as supplementary files.

REVIEWER COMMENTS

Reviewer #1 (Remarks to the Author):

The authors have done an excellent job responding to each comment and providing new data to support their model of ExoU intoxication. The article contains novel information suggesting a trafficking pattern of a membrane targeted phospholipase. These data significantly expand our understanding of ExoU's interaction with endocytic/secretory vesicles and provides potential targets to ameliorate its toxic activity.

Reviewer #2 (Remarks to the Author):

In this revised manuscript, the authors have made some efforts to address my criticisms. In some cases, they revised the text to correct mis-interpretations or overstatements, whereas in other cases, they attempted some new experiments. Specifically, I had hoped that the authors could provide more direct evidence, showing the trafficking of ExoU toxin to the plasma membrane together with DNAJC5. The authors claim that the suggested TIRF experiment did not work. They argue that "it is well established that cargoes of DNAJC5-positive vesicles are not delivered in the extracellular milieu when DNAJC5 is mutated on L115/L116". However, I am not aware of any published work showing this. The authors cited Ref. 31, 34, 35 on line 305, but these papers are clinical studies reporting the identification of these disease-associated mutations. They also mentioned ref. 52 (line 249). This paper did characterize the L115R and L116delta mutants in cell lines, but no evidence suggests that these mutations affect protein secretion. The statement "When expressed in DNAJC5^{-/-} cells, DNAJC5L115R and DNAJC5ΔL116 localized in perinuclear vesicles, but not in vesicles at the cellular periphery (Supplementary Fig. 7), a feature previously reported in PC12 neuroblastic cells 52" is also inaccurate. The cited paper actually reached the opposite conclusion that these mutants are localized in more dispersed vesicles while WT DNAJC5 is more peri-nuclear. Along the same line, another study published recently showed that DNAJC5 disease mutants show more lysosomal localization than the WT protein (Imler E. et al., 2019 eLife). Thus, while the observation that disease-associated DNAJC5 mutants fail to rescue ExoU-induced cytotoxicity in DNAJC5 null cells is interesting, and the authors' model is probably correct given the co-localization of ExoU with DNAJC5-positive vesicles and the well-established role of DNAJC5 in protein secretion, this part of the study needs to be revised to avoid ambiguity and inaccuracy as mentioned above.

The following minor points should also be addressed.

1. In line 185, "ExoU was associated with DNAJC5...." is inaccurate. There is no evidence for a physical interaction of the two proteins.
2. Line 83 should be changed to "misfolding-associated protein secretion" and the correct citation is Lee J. et al., NCB 2016.
3. Line 88-90, the papers cited here merely show the identification of DNAJC5 mutations in CLN patients. There is no evidence to suggest "alteration in MAPS has been linked to several neurological disorders". In fact, the physiological significance of the MAPS pathway remains to be established.

Reviewer #3 (Remarks to the Author):

The concerns of this reviewer have been addressed to some extent.

A limitation remains as regards the use of a hsp driver to knock-down CSP: formally, it cannot be excluded that the effects observed during the infection result from an indirect developmental mechanism, an issue absolutely not dealt with the PBS injection, a control that checks that the flies remain healthy during the experiment but does not address the response to ExoU. This formal possibility should be mentioned in the text as the experiment was not redone using other drivers that would have allowed to discard this possibility.

The results obtained with PA14 are interesting as they highlight the difference in the importance of specific virulence factors between strains and it will certainly be interesting to determine in the future the differences between PP34 and PA14 that underlie this different behavior. As many investigators work with PA14, this experiment should definitely be included in the manuscript in Supplementary Figure 4.

Minor point:

The color code used in Fig.S2b and Fig. S4c is not optimal:

- it was difficult to determine the difference between pink and purple on the printed version of i- the figure observed using artificial light
- ii-the two shades of grey for Luc-KD between PP34 DeltaexoU and PP34 are not that easy to discriminate.

Response to Reviewers' comments

Reviewer #1 (Remarks to the Author):

The authors have done an excellent job responding to each comment and providing new data to support their model of ExoU intoxication. The article contains novel information suggesting a trafficking pattern of a membrane targeted phospholipase. These data significantly expand our understanding of ExoU's interaction with endocytic/secretory vesicles and provides potential targets to ameliorate its toxic activity.

We thank Reviewer 1 for the positive comments on our work.

Reviewer #2 (Remarks to the Author):

In this revised manuscript, the authors have made some efforts to address my criticisms. In some cases, they revised the text to correct mis-interpretations or overstatements, whereas in other cases, they attempted some new experiments. Specifically, I had hoped that the authors could provide more direct evidence, showing the trafficking of ExoU toxin to the plasma membrane together with DNAJC5. The authors claim that the suggested TIRF experiment did not work. They argue that "it is well established that cargoes of DNAJC5-positive vesicles are not delivered in the extracellular milieu when DNAJC5 is mutated on L115/L116". However, I am not aware of any published work showing this. The authors cited Ref. 31, 34, 35 on line 305, but these papers are clinical studies reporting the identification of these disease-associated mutations.

They also mentioned ref. 52 (line 249). This paper did characterize the L115R and L116delta mutants in cell lines, but no evidence suggests that these mutations affect protein secretion.

We agree that there is no work reporting the loss of protein delivery when DNAJC5 is mutated on L115/L116. This was indeed an overstatement, although it is likely, (i) because of the well-known role of DNAJC5 in protein delivery (Fontaine et al, Ref 26), (ii) because of the intracellular accumulation of neurodegenerative proteins in the brain of patients bearing these mutations (reviewed in Burgoyne and Morgan, Ref 42) and (iii) as vesicles with mutated DNAJC5 do not reach the plasma membrane (Greaves et al, Ref 52, Noskova et al, Ref 34, now Ref 50).

On line 305, we wrote "ExoU toxicity was altered by mutations known to alter DNAJC5 function in vesicle trafficking, i.e., L115R and Δ L116^{31,34,35,52}". Ref 31 and 35 are indeed clinical papers that we removed from this location. Refs 34 (now 50) and 52 from Noskova et al and Greaves et al both describe the alteration of vesicle trafficking when bearing the mutated DNAJC5 (see below). Therefore, we think that Refs 34 (now 50) and 52 are relevant in this sentence. We added a review from Burgoyne and Morgan, published in 2015 (Ref 42), that recapitulates different aspects of DNAJC5 function, including vesicle trafficking.

The statement "When expressed in DNAJC5-/- cells, DNAJC5L115R and DNAJC5 Δ L116 localized in perinuclear vesicles, but not in vesicles at the cellular periphery (Supplementary Fig. 7), a feature previously reported in PC12 neuroblastic cells 52" is also inaccurate. The cited paper actually reached the opposite conclusion that these mutants are localized in more dispersed vesicles while WT DNAJC5 is more peri-nuclear.

In Ref 52, Greaves et al wrote "*Wild-type EGFP-CSP associates with the plasma membrane and vesicles in PC12 cells (16) (Fig. 1B, left). In contrast, both the L116 and L115R mutants displayed a more dispersed and punctate localization and a reduced plasma membrane staining (Fig. 1B, middle and right).*" The images shown by these authors in Fig. 1B are very similar to what we observed in A549 cells, and we think it is important to mention this similarity in the text. We modified the text to focus on the fact that vesicles remain intracellular: "When expressed in DNAJC5^{-/-} cells, DNAJC5^{L115R} and DNAJC5^{ΔL116} localized in **intracellular** vesicles, but **not at the plasma membrane** (Supplementary Fig. 7)".

Importantly, Noskova et al (Ref 34, now 50) also report that vesicles bearing the mutated DNAJC5 are not observed at the plasma membrane: "*Using immunofluorescence analysis, we found wild-type EGFP-CSP α predominantly at the plasma membrane, whereas both mutated proteins showed diffuse intracellular staining and abnormal colocalization with markers for the ER and Golgi apparatus (Figure 4A).*"

Along the same line, another study published recently showed that DNAJC5 disease mutants show more lysosomal localization than the WT protein (Imler E. et al., 2019 eLife).

Imler et al (2020) provide a very detailed study on the trafficking of pathological CSP mutants in *Drosophila*. Although this study is interesting and important, we did not quote this article because our work is mainly focused on ExoU targeting at the plasma membrane via DNAJC5 vesicles, and not on the cellular and molecular description of this pathology.

Thus, while the observation that disease-associated DNAJC5 mutants fail to rescue ExoU-induced cytotoxicity in DNAJC5 null cells is interesting, and the authors' model is probably correct given the co-localization of ExoU with DNAJC5-positive vesicles and the well-established role of DNAJC5 in protein secretion, this part of the study needs to be revised to avoid ambiguity and inaccuracy as mentioned above.

We hope that the modifications provided in this new version bring more accuracy to our paper.

The following minor points should also be addressed.

1. In line 185, "ExoU was associated with DNAJC5...." is inaccurate. There is no evidence for a physical interaction of the two proteins.

The sentence was modified to "ExoU **colocalized** with DNAJC5..."

2. Line 83 should be changed to "misfolding-associated protein secretion" and the correct citation is Lee J. et al., NCB 2016.

These modifications were made in the new version.

3. Line 88-90, the papers cited here merely show the identification of DNAJC5 mutations in CLN patients. There is no evidence to suggest "alteration in MAPS has been linked to several

neurological disorders". In fact, the physiological significance of the MAPS pathway remains to be established.

The sentence has been suppressed.

Reviewer #3 (Remarks to the Author):

The concerns of this reviewer have been addressed to some extent. A limitation remains as regards the use of a hsp driver to knock-down CSP: formally, it cannot be excluded that the effects observed during the infection result from an indirect developmental mechanism, an issue absolutely not dealt with the PBS injection, a control that checks that the flies remain healthy during the experiment but does not address the response to ExoU. This formal possibility should be mentioned in the text as the experiment was not redone using other drivers that would have allowed to discard this possibility.

Drosophila Csp mutants rapidly die after birth from neurological disorders as shown in Zinsmaier, K. E., Eberle, K. K., Buchner, E., Walter, N. & Benzer, S. Paralysis and early death in cysteine string protein mutants of Drosophila. Science 263, 977-980, doi:10.1126/science.8310297 (1994). If a putative leaky expression of the hsp driver used to knock down CSP would occur and result in developmental defects, as suggested by Reviewer 3, the CSP-KD flies should display a phenotype reminiscent to the one observed in Csp mutants. It is not the case, as CSP-KD flies did not display any evidence of neurological disorders and early death (not shown). They are healthy enough to deal normally with PBS injection and are even more resistant to ExoU than control flies. However, we included this possibility in the Results at lines 161-163: "Although one cannot formally exclude an unlikely developmental effect (see Materials and Methods for details), these data indicate that DNAJC5/CSP is required for full ExoU-dependent P. aeruginosa virulence in vivo." and in the Materials and Methods at lines 452-456: "Although one cannot formally rule out the possibility that a leaky expression of the hs-Gal4 driver occurs during development and results in developmental csp knock-down, the fact that the CSP-KD flies did not display any evidence of neurological disorders and early death (not shown), as observed in csp mutants, strongly argues against it."

The results obtained with PA14 are interesting as they highlight the difference in the importance of specific virulence factors between strains and it will certainly be interesting to determine in the future the differences between PP34 and PA14 that underlie this different behavior. As many investigators work with PA14, this experiment should definitely be included in the manuscript in Supplementary Figure 4.

This experiment was included in Supplementary Fig. 4 and commented in the text: "Interestingly, the survival curves of *Drosophila* infected with PA14 or PA14 Δ exoU were not significantly different (Supplementary Fig. 4e), suggesting that PA14 lethal effect in *Drosophila* is mostly mediated by virulence factors other than ExoU. Therefore, PA14 was not used to establish the role of CSP in this in-vivo assay to examine the role of CSP in ExoU toxicity. "

Minor point:

The color code used in Fig.S2b and Fig. S4c is not optimal:

-it was difficult to determine the difference between pink and purple on the printed version of

i- the figure observed using artificial light

ii-the two shades of grey for Luc-KD between PP34 DeltaexoU and PP34 are not that easy to discriminate.

The color codes have been changed in the two figures to be more different.

REVIEWERS' COMMENTS

Reviewer #2 (Remarks to the Author):

The authors have addressed all my criticisms.

Reviewer #3 (Remarks to the Author):

The authors have addressed the concerns of this reviewer.

Of note, typo line 164 on Drosophila; Fig. S4a: UAS-siRNA and not USA-siRNA

Response to Reviewers' comments

Reviewer #2 (Remarks to the Author):

The authors have addressed all my criticisms.

Reviewer #3 (Remarks to the Author):

The authors have addressed the concerns of this reviewer.

Of note, typo line 164 on Drosophila; Fig. S4a: UAS-siRNA and not USA-siRNA

The typo line 164 on Drosophila has been corrected

USA was changed to UAS in Fig. S4a